# ⚑ DeepEyesV2: Toward Agentic Multimodal Model

**Jack Hong**[*], **Chenxiao Zhao**[*], **ChengLin Zhu**[*], **Weiheng Lu**, **Guohai Xu**[†], **XingYu**
Xiaohongshu Inc.
`jaaackhong@gmail.com,{chenxiao2, xuguohai}@xiaohongshu.com`

Project Page: `https://visual-agent.github.io/`

## ABSTRACT

Agentic multimodal models should not only comprehend text and images, but also actively invoke external tools, such as code execution environments and web search, and integrate these operations into reasoning. In this work, we introduce *DeepEyesV2* and explore how to build an agentic multimodal model from the perspectives of data construction, training methods, and model evaluation. We observe that direct reinforcement learning alone fails to induce robust tool-use behavior. This phenomenon motivates a two-stage training pipeline: a cold-start stage to establish tool-use patterns, and reinforcement learning stage to further refine tool invocation. We curate a diverse, moderately challenging training dataset, specifically including examples where tool use is beneficial. We validate *DeepEyesV2* across real-world understanding, mathematical reasoning, and search-intensive benchmarks, demonstrating that systematic tool integration enables reliable and extensible multimodal reasoning behaviour. Moreover, *DeepEyesV2* exhibits task-adaptive tool invocation, tending to use image operations for perception tasks and numerical computations for reasoning tasks. Reinforcement learning further enable complex tool combinations and allowing model to selectively invoke tools based on problem context. We hope our study can provide guidance for community in developing agentic multimodal models.

## 1 INTRODUCTION

An agentic multimodal model should not only be capable of understanding text and images, but can also actively invoke tools (*e.g.*, a code execution environment or a web search interface) and seamlessly integrate these operations into its advanced reasoning process. For example, as illustrated in Figure 1, when asked for the mean value of the black dots in the figure, agentic multimodal model first crops each subplot to inspects the numeric values for each black point, and then emits executable Python code to compute the average values. Likewise, when a query requires a stock quote, agentic multimodal model can issue image- or text-based search queries and invokes an external API to obtain the quote. By integrating programmatic image operations, computation, and retrieval into a single reasoning loop, these models enable fine-grained visual perception, support complex mathematical reasoning, and continually incorporate updated external knowledge, thus achieving more reliable and comprehensive multimodal inference.

Although existing multimodal models demonstrate strong perception and interpretation abilities, they remain largely passive and lack the ability to autonomously invoke external tools, which is essential for agentic multimodal models. These tool-use capabilities can be categorized into two types: (i) Operation tools: Current models cannot perform complex operations on visual or numerical data, including fine-grained image manipulations (*e.g.*, cropping, measuring) and quantitative computations. This limits their capacity to reason about detailed visual content or solve mathematical problems. (ii) Information retrieval tools: Models cannot proactively access up-to-date external knowledge, which often leads to outdated conclusions or statements without verifiable sources.

---

[*]Equal Contribution.
[†]Corresponding author.

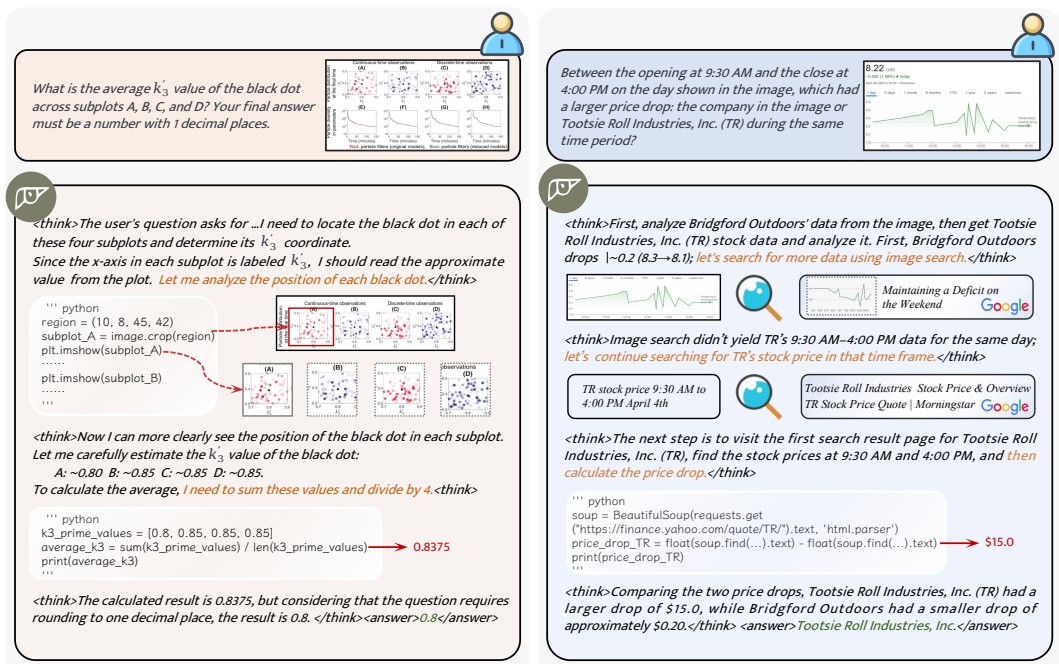

Figure 1: **Case reasoning trajectory of *DeepEyesV2*.** *DeepEyesV2* seamlessly integrates code execution and web search within its iterative reasoning process. Notably, in the right case, the behavior of accessing webpages via code does not exist in cold start data and is spontaneously acquired during reinforcement learning.

While o3 (OpenAI, 2025) has explored "thinking with image" reasoning pattern, how to realize such capabilities remains unclear. Existing approaches are either restricted to perception tasks without support for complex reasoning or web search (Zhang et al., 2025b), or rely on limited tools (*e.g.* (Zheng et al., 2025b)), leaving a substantial gap to truly agentic multimodal models.

To explore how to construct such agentic multimodal models, we introduce *DeepEyesV2*, which seamlessly integrates tool invocation within the dynamic reasoning loop. *DeepEyesV2* actively decides when and how to invoke tools, enabling a dynamic process of evidence acquisition and verification. Then, tool outputs are iteratively incorporated into reasoning process, allowing model to refine its hypotheses, validate intermediate results, and ultimately arrive at more reliable and interpretable conclusions. In this work, we systematically investigate key aspects of building an agentic MLLM, including model training strategies, dataset curation, and evaluation protocols.

We first follow the setup of DeepEyes (Zheng et al., 2025b) and apply reinforcement learning directly on Qwen2.5-VL (Bai et al., 2025), but find that limited inherent tool-use capability prevents stable tool invocation. This highlights the need for a cold-start stage to establish reliable tool-use patterns. Thus, we curate a high-quality dataset that spans diverse scenarios, including perception, reasoning, and search tasks. After cleaning, we apply two filters: (i) difficulty filtering, retaining only questions unsolvable by the base model, and (ii) tool-benefit classification, keeping cases where tool use improves accuracy. Data are split into two subsets: tool-solvable examples for RL and harder unsolved cases for cold start, further augmented with long chain-of-thought trajectories. Supervised fine-tuning on this cold-start dataset enables the model to acquire basic tool-use patterns and deeper reasoning, after which RL further strengthens tool invocation. Notably, we rely only on two simple rewards, accuracy and format, without complex reward engineering (Su et al., 2025).

We assess *DeepEyesV2* on benchmarks spanning real-world understanding, mathematical reasoning, and search-intensive tasks. *DeepEyesV2* outperforms both general-purpose MLLMs and prior specific reasoning approaches. Specifically, on real-world understanding benchmarks, *DeepEyesV2* surpasses even Qwen2.5-VL-32B in some benchmarks through effective tool use. On reasoning tasks, *DeepEyesV2* shows performance gains across multiple benchmarks, including +7.1 on Math-Verse (52.7% accuracy). On search benchmarks, *DeepEyesV2* delivers strong advantages, reaching 63.7% on MMSearch (Jiang et al., 2024), far beyond the MMSearch-R1 (Wu et al., 2025) (53.8%).

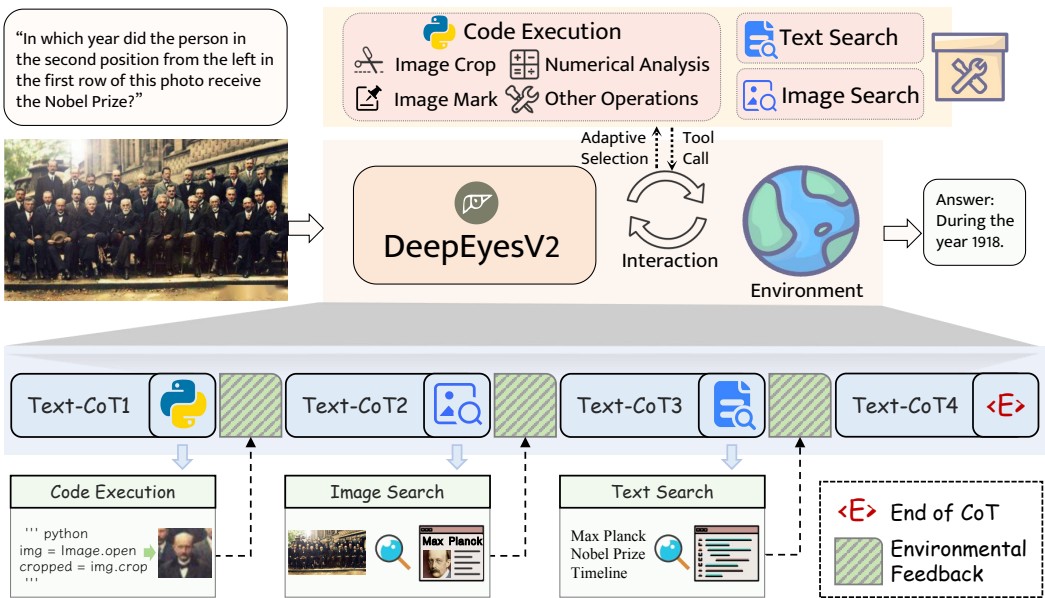

Figure 2: **Pipeline of *DeepEyesV2*.** *DeepEyesV2* invokes tools and incorporates execution results into subsequent reasoning steps, enabling iterative and tool-augmented multimodal inference.

These results demonstrate that by reliably invoking tools, *DeepEyesV2* extends its comprehensive capabilities, achieving accurate and advanced reasoning.

We observe the task-dependent tool invocation patterns in *DeepEyesV2*. For perception tasks, *DeepEyesV2* primarily uses image operations, such as cropping, to extract fine-grained visual details, whereas for reasoning tasks, *DeepEyesV2* favors numerical analysis. Moreover, reinforcement learning can further enhances tool-use behavior, enabling more complex tool combinations and adaptive decision-making. *DeepEyesV2* learns to selectively invoke tools based on the problem context, reflecting the emergence of autonomous, agentic reasoning.

The main contributions are summarized as follows: (i) We introduce *DeepEyesV2*, an agentic multimodal model that unifies code execution and web search within a single reasoning loop, enabling reliable and complex reasoning. (ii) We construct a carefully curated training corpus through rigorous data filtering and cleaning. The resulting dataset is diverse in task types, of appropriate difficulty, and explicitly designed to ensure the beneficial integration of tools. Based on this, we build both cold-start SFT data and RL data that complement each other. (iii) Extensive experiments across real-world understanding, mathematical reasoning, and search-intensive benchmarks demonstrate the strong reasoning and tool-usage ability of *DeepEyesV2*. (iv) We analyze the dynamics of tool-use behavior in *DeepEyesV2*, revealing task-adaptive patterns. Besides, we also find reinforcement learning can enable more complex tool combinations and adaptive, context-aware tool invocation.

## 2 RELATED WORKS

**Multimodal Large Language Models.** The field of multimodal large language models (MLLMs) has witnessed rapid progress in recent years. Early efforts mainly focus on combining pretrained visual encoders with large language models through lightweight adapters or projection layers, enabling basic vision–language alignment and simple multimodal understanding (Li et al., 2023; Liu et al., 2023; 2024a; Bai et al., 2023; Chen et al., 2024). Subsequently, more powerful architectures such as Qwen2.5-VL (Bai et al., 2025), LLaVA-OneVision (Li et al., 2024a), and InternVL3 (Zhu et al., 2025), expand the training scale and integrated more diverse visual data, significantly improving performance on benchmarks of visual question answering, captioning, and general perception tasks. Recently, some OmniMLLMs (Li et al., 2025b; Zhao et al., 2025a; Fu et al., 2024; Jain et al., 2024; Hong et al., 2025) are capable of processing a mix of modalities like speech, video, and images simultaneously. However, existing MLLMs remain largely passive: they can interpret multimodal

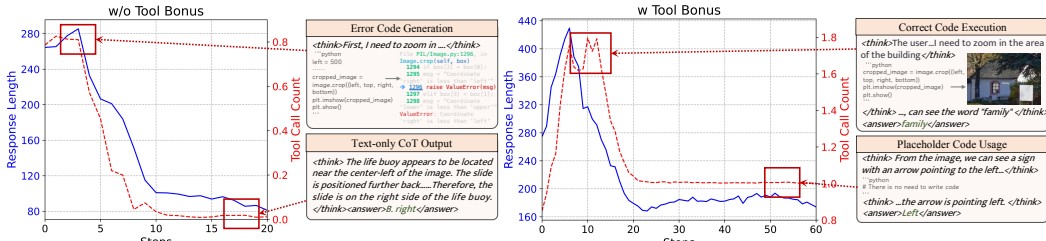

Figure 3: **Pioneer Experiments** reveal that existing multimodal models cannot directly acquire reliable tool use ability through RL, demonstrating the necessity of a cold start phase.

inputs and generate answers, but lack the ability to actively invoke external tools for computation or knowledge retrieval, which limits their reliability in complex reasoning tasks.

**Thinking with Images.** The paradigm of "Think with Image" is first introduced by o3 (OpenAI, 2025), which demonstrated that multimodal models can interleave reasoning with iterative visual analysis, actively manipulating images to support step-by-step problem solving. Many works attempt to reproduce such capabilities. Most approaches (Su et al., 2025; Lai et al., 2025; Liu et al., 2025; Fan et al., 2025; Jiang et al., 2025; Zhang et al., 2025a) adopt a two-stage training pipeline, where a cold-start phase is followed by reinforcement learning. In contrast, DeepEyes (Zheng et al., 2025b) only adopts reinforcement learning alone and incentivizes the "Think with Image" behaviors, leading to strong reasoning performance. However, the majority of these efforts employ a rather limited tool set, typically restricted to region cropping for fine-grained perception. To improve generality, PyVision (Zhao et al., 2025b) and Thyme (Zhang et al., 2025b) utilize code execution to enable more flexible visual operations. Despite this progress, these models remain constrained to image manipulation only, and are unable to handle knowledge-intensive questions where access to up-to-date external information is essential.

**Search-oriented Reasoning.** To mitigate the inherent knowledge limitations of large multimodal language models, a growing line of work explores augmenting them with external knowledge acquisition. Early approaches commonly adopt the retrieval-augmented generation (RAG) paradigm (Song et al., 2025; Jin et al., 2025), where relevant information is retrieved from a pre-constructed knowledge base and fed into the model. While effective, this paradigm remains constrained by the static and finite nature of the underlying corpus. To overcome these limitations, more recent studies attempt to leverage online search to dynamically access broader and up-to-date information (Zheng et al., 2025a). Beyond purely textual queries, some efforts extend search into the multimodal domain, enabling retrieval of not only documents but also images, charts, or other media forms relevant to the task (Wu et al., 2025; Wang et al., 2025d). These advances highlight the potential of search-augmented reasoning to complement perception and tool-use capabilities, ultimately broadening the scope of problems that multimodal models can effectively address.

## 3 *DeepEyesV2*

We explore how to construct agentic multimodal models from the perspectives of training strategy, dataset design, and evaluation. We begin in Section 3.1 by presenting the overall pipeline of *DeepEyesV2*, which integrates tool invocation into the reasoning loop. Then, we conduct pioneer experiments in Section 3.2 to reveal the limitations of existing models in reliably using tools, underscoring the necessity of a cold-start stage. After that, we describe the curation of a high-quality training dataset and the principles behind its construction in Section 3.3. Finally, building on the cold-start foundation, we apply a reinforcement learning stage to further enhance the efficiency and flexibility of tool-use behavior, which is described in Section 3.4.

### 3.1 OVERALL PIPELINE

Similar to DeepEyes (Zheng et al., 2025b), *DeepEyesV2* is an agentic multimodal model, but with extended tool-use capabilities beyond simple cropping. In *DeepEyesV2*, programmatic code execution and web retrieval are treated as complementary and interleavable tools inside a single reasoning trajectory (see Figure 2). Given an image input and the corresponding user query, *DeepEyesV2* first generates an initial reasoning plan, and explicitly determines whether this question can be solved

Table 1: **Results on real-world & OCR & chart understanding Benchmarks.**

| Model | Tool | Param Size | Real-World Understanding | | | | | OCR | | Chart | | |
|---|---|---|---|---|---|---|---|---|---|---|---|---|
| | | | V* Bench | HRBench 4K | HRBench 8K | MME-RealWorld | Tree Bench | OCR Bench | SEED 2 Plus | CharXiv descriptive | CharXiv reasoning | Chart QA |
| *Open-source Models* | | | | | | | | | | | | |
| LLaVA-OV | ✗ | 7B | 75.4 | 63.0 | 59.8 | 57.4 | 37.3 | - | - | - | - | 80.0 |
| Qwen2.5-VL | ✗ | 7B | 78.5 | 71.6 | 67.9 | 57.3 | 37.0 | 864 | 70.4 | 72.7 | 40.2 | 86.2 |
| Qwen2.5-VL | ✗ | 32B | 80.6 | 74.1 | 69.9 | - | 42.5 | - | **72.4** | **83.2** | 48.0 | - |
| InternVL3 | ✗ | 8B | 81.2 | 70.0 | 69.3 | - | 38.8 | 880 | 69.7 | 73.6 | 37.6 | 86.6 |
| *Grounded Reasoning Models* | | | | | | | | | | | | |
| Pixel-Reasoner | Crop | 7B | 84.3 | 74.0 | 66.9 | 64.4 | 39.0 | - | - | - | - | - |
| DeepEyes | Crop | 7B | **85.6** | 75.1 | 72.6 | - | 37.5 | - | - | - | - | - |
| Thyme | Code | 7B | 82.2 | 77.0 | 72.0 | 64.8 | - | 863 | - | - | - | 86.1 |
| *Agentic Multimodal Model* | | | | | | | | | | | | |
| *DeepEyesV2* | General | 7B | 81.8 | **77.9** | **73.8** | 64.9 | 42.5 | **882** | 70.5 | 78.6 | **48.9** | **88.4** |
| Δ (*vs* Qwen2.5-VL-7B) | | | +3.3 | +6.3 | +5.9 | +7.6 | +5.5 | +18 | +0.1 | +5.9 | +8.7 | +2.2 |

directly through internal reasoning or requires tool invocation. If tool use is necessary, *DeepEyesV2* emits executable Python code or issues web search queries. Code execution is carried out in a sandboxed environment and can produce structured outputs such as transformed images, numerical measurements, computed arrays, plots, or execution logs. Image queries are submitted via SerpAPI and return the top five visually matched webpages (each with a thumbnail and title). Text queries return the five most relevant webpages, along with titles and snippets. All tool outputs are converted into observations and appended to model's context. *DeepEyesV2* then thinks further in light of these observations and may plan further tool invocations (either additional code, further searches, or both), iterating this reasoning–tooling–integration loop until a conclusive answer is produced.

*DeepEyesV2* can dynamically choose, combine, and use tools as reasoning unfolds. This integration yields three main advantages: (i) it allows **expanded and enhanced analytical capability** through executable code; (ii) it enables **active and real-time knowledge seeking** by retrieving multimodal evidence from the web; and (iii) it supports **iterative, interleaved multi-tool reasoning**, in which code execution and search can be dynamically combined within a single trajectory, rather than being isolated modules. Together, these features position *DeepEyesV2* as a more general, reliable, and extensible framework for multimodal reasoning.

## 3.2 PIONEER EXPERIMENTS

To investigate whether MLLMs can directly acquire tool-use ability through reinforcement learning, we first conduct a pioneer experiment on Qwen2.5-VL (Bai et al., 2025) following DeepEyes (Zheng et al., 2025b). As shown in Figure 3, during training, we observe that in the early stages model occasionally attempts to produce Python code, but these outputs are often buggy or fail to execute, indicating that existing MLLMs struggle to generate stable and reliable code. As training continues, model gradually abandons code generation and converges to producing only short reasoning chains followed by direct answers, thereby bypassing tool use. Then, to encourage tool invocation, we incorporate the tool usage bonus mechanism from DeepEyes, which explicitly rewards the generation of code. With this additional signal, model is indeed able to produce correct and runnable code in the early stages, suggesting that the mechanism can enforce coding ability. However, with continued training a new degeneration emerges: model's behavior converged to emitting exactly one code block per query, and this single block typically consists of non-executable, placeholder comments rather than meaningful code, revealing the phenomenon of reward hacking. This pioneer experiment highlights that existing MLLMs cannot reliably learn complex tool use through direct RL alone, motivating the need for a cold start to bootstrap model's tool invocation ability.

## 3.3 DATA CURATION

**Data Collection.** Pioneer experiments have highlighted the necessity of constructing a high-quality dataset for supervised fine-tuning to explicitly guide model to learn how to generate executable code and perform tool invocations. Following DeepEyes (Zheng et al., 2025b), we collect data in accordance with the following principles: (i) **Diverse tasks and image distribution.** We incorporate varied data to cover a wide range of multimodal challenges and visual components. (ii) **Verifiability and structured format.** All questions are reformulated into a structured, open-ended QA format

Table 2: **Results on multimodal reasoning benchmarks.**

| Model | Tool | Param Size | MathVista | MathVerse | MathVision | WeMath | DynaMath | LogicVista |
|---|---|---|---|---|---|---|---|---|
| | | | *Open-source Models* | | | | | |
| LLaVA-OV | ✗ | 7B | 58.6 | 19.3 | 18.3 | 20.9 | - | 33.3 |
| Qwen-2.5-VL | ✗ | 7B | 68.3 | 45.6 | 25.6 | 34.6 | 53.3 | 45.9 |
| InternVL3 | ✗ | 8B | 71.6 | 39.8 | 29.3 | 37.1 | - | 44.1 |
| | | | *Text-only Reasoning Models* | | | | | |
| MM-Eureka | ✗ | 7B | 72.6 | - | 28.1 | 21.8 | - | 46.3 |
| ThinkLite | ✗ | 7B | 71.6 | - | 24.6 | **41.8** | - | 42.7 |
| VL-Rethinker | ✗ | 7B | **73.7** | - | 28.4 | 36.3 | - | 42.7 |
| VLAA-Thinker | ✗ | 7B | 71.7 | - | 24.2 | 35.7 | - | 45.9 |
| | | | *Grounded Reasoning Models* | | | | | |
| DeepEyes | Crop | 7B | 70.1 | 47.3 | 26.6 | 38.9 | 55.0 | 47.7 |
| Thyme | Code | 7B | 70.0 | - | 27.6 | 39.3 | - | **49.0** |
| | | | *Agentic Multimodal Model* | | | | | |
| *DeepEyesV2* | General | 7B | 71.9 | **52.7** | **28.9** | 38.1 | **57.2** | 48.7 |
| Δ (*vs* Qwen2.5-VL-7B) | | | +3.6 | +7.1 | +3.3 | +3.5 | +3.9 | +2.8 |

Table 3: **Results on search-oriented benchmarks.**

| Model | Tool | Model Size | FVQA-test | InfoSeek | MMSearch | SimpleVQA |
|---|---|---|---|---|---|---|
| | | | *Open-source & Proprietary Models* | | | |
| GPT4o | ✗ | - | 41.7 | 42.7 | 22.2 | 46.6 |
| Gemini 2.5 Pro | ✗ | - | 37.2 | 37.0 | 26.9 | 53.4 |
| Qwen-2.5-VL | ✗ | 7B | 20.3 | 20.1 | 12.8 | 38.4 |
| | | | *Search Models* | | | |
| Qwen-2.5-VL | Search | 7B | 52.9 | 53.7 | 52.2 | 51.6 |
| MMSearch-R1 | Search | 7B | 58.4 | **55.1** | 53.8 | 57.4 |
| WebWatcher | Search | 7B | - | - | 49.1 | 54.3 |
| | | | *Agentic Multimodal Model* | | | |
| *DeepEyesV2* | General | 7B | **60.6** | 51.1 | **63.7** | **59.4** |
| Δ (*vs* Qwen2.5-VL-7B Search) | | | +7.7 | -2.6 | +11.5 | +7.8 |

to facilitate objective evaluation. We exclude examples that cannot be reliably verified, such as those with incorrect answers, ambiguous phrasing, or poor readability. (iii) **Appreciate difficulty.** We exclude examples that the base model can easily solve and prioritize questions that remain challenging. (iv) **Beneficial integration of tools.** We categorize examples based on whether tool usage leads to correct answers. Cases where model can solve correctly using additional tool calls are reserved for reinforcement learning, whereas examples that remain unsolved even with tool assistance are used for cold start.

Specially, we curate data from three major categories: perception, reasoning, and search. Besides, we also include long Chain-of-Cot (CoT) reasoning data in cold start subset. Please refer to Appendix A.2 for more details on data sources. All datasets are carefully cleaned, reformatted, and divided into subsets for cold start or reinforcement learning subsets. To ensure sufficient difficulty, we employ Qwen2.5-VL-7B (Bai et al., 2025) as a baseline evaluator. For each question, model is prompted to generate 8 responses, and we retain only those instances where it answers correctly at most two times, thereby filtering out trivial cases. To further assess tool-use effectiveness, we prompt model to solve each question with tool invocation, again collecting 8 responses per instance, and categorize examples according to their success rate.

**Trajectories Synthesis.** We construct cold start datasets by eliciting step-by-step trajectories from models (*e.g.*, Gemini 2.5 Pro (Comanici et al., 2025), GPT-4o (Hurst et al., 2024), and Claude Sonnet 4 (Anthropic, 2025)). For each prompt, these models are prompted to produce detailed reasoning traces that explicitly include tool-invocation markers (*e.g.*, code snippets). Each declared tool call is executed, and the returned outputs are fed back to the originating model, and model continues reasoning, potentially issuing further tool calls, until it produces a final answer. The entire

Table 4: **Ablation study on cold start data**. Long CoT refers to text-only reasoning data. For more details about cold start data, please refer to Appendix A.2.

| Perception | Reason | Long CoT | V* Bench | SEED 2 Plus | CharXiv descriptive | CharXiv reasoning | Math Vista | Math Verse |
|:---:|:---:|:---:|:---:|:---:|:---:|:---:|:---:|:---:|
| | Qwen-2.5-VL-7B | | 63.9 | 69.2 | 68.9 | 35.7 | 65.3 | 36.2 |
| ✓ | | | 78.0 | 68.2 | 70.6 | 40.8 | 66.8 | 38.4 |
| | ✓ | | 76.9 | 66.3 | 68.1 | 38.7 | 63.6 | 36.7 |
| | ✓ | ✓ | 75.9 | 68.7 | 72.0 | 43.1 | 68.2 | 47.6 |
| ✓ | ✓ | ✓ | 78.5 | 69.6 | 73.4 | 44.3 | 68.3 | 47.1 |

Table 5: **Ablation study on reinforcement learning data**. DeepEyesV2-SFT denotes the model after cold start. For more details about reinforcement learning data, please refer to Appendix A.2.

| Perception | Reason | Search | V* Bench | SEED 2 Plus | CharXiv descriptive | CharXiv reasoning | Math Vista | Math Verse | Info Seek | MM Search |
|:---:|:---:|:---:|:---:|:---:|:---:|:---:|:---:|:---:|:---:|:---:|
| | *DeepEyesV2*-SFT | | 78.5 | 69.6 | 73.4 | 44.3 | 68.3 | 47.1 | 47.9 | 56.8 |
| ✓ | | | 79.3 | 70.2 | 76.0 | 45.6 | 69.5 | 47.6 | 44.6 | 52.6 |
| | ✓ | | 77.4 | 69.3 | 72.3 | 45.2 | 70.4 | 49.8 | 43.0 | 53.7 |
| ✓ | ✓ | | 80.9 | 70.4 | 78.2 | 48.7 | 71.2 | 52.0 | 44.2 | 55.0 |
| ✓ | ✓ | ✓ | 81.8 | 70.5 | 78.6 | 48.9 | 71.9 | 52.7 | 51.1 | 63.7 |

interaction is recorded as a single trajectory. Only trajectories with correct final answers and error-free code are retained, ensuring high-quality cold-start data.

## 3.4 AGENTIC REINFORCEMENT LEARNING

After cold-start training has equipped model with basic tool-use patterns, we adopt reinforcement learning to further enhance its ability to integrate tools in dynamic environment. Unlike SFT, which relies on learning from static trajectories, agentic RL places the model in an interactive environment where it must dynamically decide when and how to invoke tools in order to solve tasks. Following DeepEyes (Zheng et al., 2025b), we employ a sparse and outcome-driven reward. The overall reward consists of two components: (i) accuracy reward $R_{acc}$, which evaluates whether the final answer matches the ground truth, and (ii) format reward $R_{format}$, which penalizes outputs that violate required formats. The total reward is defined as: $R = R_{acc} + R_{format}$.

## 4 EXPERIMENTS

### 4.1 IMPLEMENTATION DETAILS

We conduct training in two stages: cold start SFT and reinforcement learning. The backbone model is Qwen2.5-VL-7B (Bai et al., 2025). For SFT, we train with a batch size of 128 and a learning rate of $1 \times 10^{-5}$. Model is optimized for 3 epochs using AdamW (Loshchilov & Hutter, 2017) optimizer with cosine learning rate decay. For RL, we adopt DAPO (Yu et al., 2025) as the optimization algorithm, with a batch size of 256 and 16 rollouts per prompt. The KL coefficient is set to 0.0, and the maximum response length is capped at $16,384$ tokens.. The learning rate is $1 \times 10^{-6}$, and the upper and lower clip ratios are 0.30 and 0.20, respectively.

### 4.2 MAIN RESULTS

**Real-World & OCR & Chart Understanding.** We evaluate *DeepEyesV2* across three categories of benchmarks: real-world understanding, OCR, and chart understanding. For comparison, we include two kinds of models: (i) open-source general-purpose MLLMs, including LLaVA-OneVision (Li et al., 2024a), Qwen2.5-VL (Bai et al., 2025), and InternVL3 (Zhu et al., 2025); and (ii) grounded reasoning models, such as DeepEyes (Zheng et al., 2025b) and Thyme (Zhang et al., 2025b). Deep-Eyes performs fine-grained perception by cropping the target region, while Thyme manipulates images through executable code. Compared to base model Qwen2.5-VL-7B, *DeepEyesV2* demonstrates substantial performance gains, and even surpasses Qwen2.5-VL-32B in some benchmarks

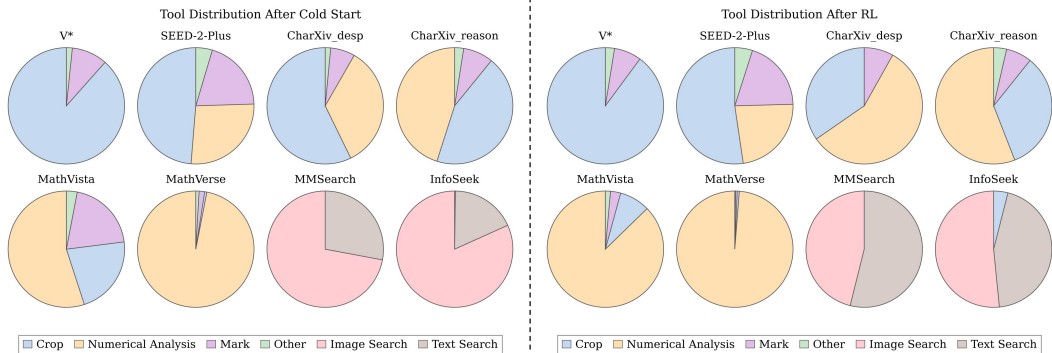

Figure 4: **Tool distribution comparison.** *DeepEyesV2* demonstrates the task-specific tool-calling distribution across different tasks.

(Table 1), highlighting the effectiveness of tool-augmented reasoning. Moreover, it consistently outperforms existing grounded reasoning models. These results indicate that dynamic tool invocation enables model to extract fine-grained details, thereby improving real-world scene comprehension.

**Multimodal Reasoning.** We further evaluate *DeepEyesV2* on mathematical reasoning benchmarks to assess its strong reasoning capability. As shown in Table 2, we compare *DeepEyesV2* against existing open-source MLLMs, such as Qwen2.5-VL (Bai et al., 2025), text-only multimodal reasoning models, including MM-Eureka (Meng et al., 2025), and grounded reasoning models, such as DeepEyes (Zheng et al., 2025b) and Thyme (Zhang et al., 2025b). *DeepEyesV2* consistently outperforms these alternatives, and notably achieves stronger results than text-only multimodal reasoning models, underscoring the benefit of tool use for enhancing mathematical reasoning.

**Online Searching.** To further examine the effectiveness of external information acquisition, we evaluate *DeepEyesV2* on search-oriented benchmarks. These datasets encompass knowledge-intensive visual question answering, fact verification, and multimodal retrieval-based reasoning, all of which require models to go beyond perceptual understanding and actively retrieve external evidence. For comparison, we benchmark *DeepEyesV2* against both general-purpose MLLMs such as Qwen2.5-VL, Gemini 2.5 Pro, and GPT4o, as well as models where search capability is incorporated (Jiang et al., 2024; Geng et al., 2025). As shown in Table 3, *DeepEyesV2* demonstrates superior search capabilities, achieving consistently higher accuracy across all benchmarks.

## 4.3 TOWARDS AGENTIC MLLM

**Training Data.** To understand how training data influences the development of tool-use ability, we investigate the impact of different dataset compositions.

• **Cold Start Data.** We conduct ablations on the SFT dataset (Table 4). Directly evaluating Qwen2.5-VL-7B brings a great performance drop and confirms that existing MLLMs lack robust tool-use ability. Training only on perception data helps perception benchmarks but not reasoning; training only on reasoning data yields limited or negative gains, showing perception and reasoning rely on distinct tool-use patterns, with reasoning being more complex and harder to master. Adding long CoT trajectories substantially enhances reasoning and tool use, demonstrating that stronger thinking ability directly facilitates better tool use. Combining perception, reasoning, and CoT data achieves the best overall results, highlighting the complementary benefits of diverse supervision and the value of long CoT for complex reasoning. Overall, these results highlight two key factors of cold start data: (i) diversity, as perception and reasoning rely on different tool-use patterns and data with diverse tasks should be involved to improve generalization; and (ii) the inclusion of long CoT data, which strengthens reasoning and substantially improves tool use on complex tasks.

• **RL Data.** We further conduct ablation studies on different subsets of RL data. Results are shown in Table 5. Using only perception or only reasoning data improves corresponding benchmarks but harms others, while combining both yields consistent gains. Adding search data further boosts retrieval benchmarks, leading to balanced overall performance. These results emphasize that data diversity is critical for reinforcement learning in agentic multimodal models.

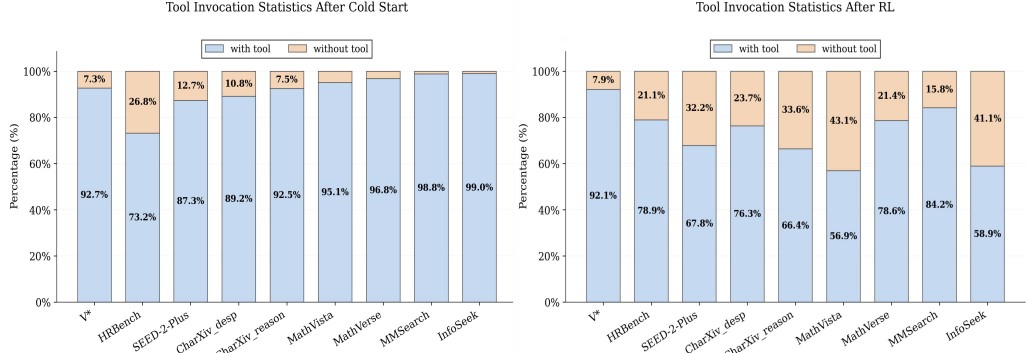

Figure 5: **Tool invocation statics.** Reinforcement Learning enhances *DeepEyesV2*'s tool-calling flexibility, allowing it to decide dynamically whether to invoke tools.

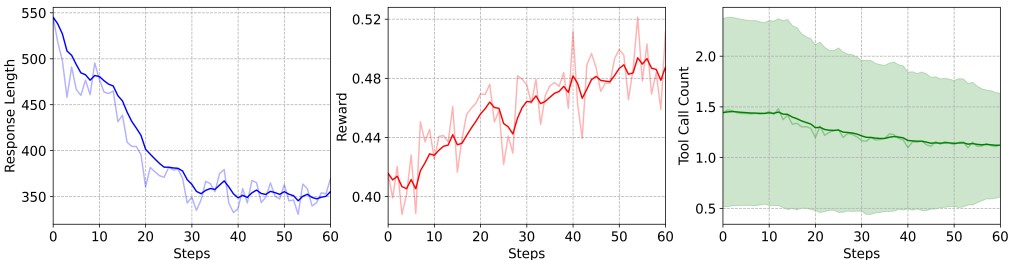

Figure 6: **Training dynamics of RL.** Reinforcement learning improves the efficiency of *Deep-EyesV2*'s reasoning and tool usage.

**In-Deep Analysis.** Then, we conduct an in-depth analysis of *DeepEyesV2*'s tool-use behavior after cold start and RL, comparing the two stages to better understand how training shapes and alters the model's strategies for invoking tools.

• **Tool Distribution.** To understand how the model leverages tools across various scenarios, we analyze tool-use distributions over eight benchmarks before and after reinforcement learning (Figure 4). *DeepEyesV2* exhibits clear task-dependent preferences: in real-world perception tasks (V*), model mainly uses cropping to obtain fine-grained visual details; in OCR tasks (SEED-Bench-2-Plus), it additionally performs region marking and numerical computations; chart-related tasks (CharXiv) involve more arithmetic operations; reasoning benchmarks (MathVista, MathVerse) are dominated by mathematical computations for intermediate verification and final answers; and search tasks (MM-Search, InfoSeek) primarily invoke search tools. Moreover, when comparing behaviors before and after RL, we observe a notable shift. After reinforcement learning, model tends to perform more numerical operations across multiple tasks, and begins to integrate image manipulation tools (*e.g.*, cropping) with search in search benchmarks, indicating that RL helps model develop a more synergistic use of heterogeneous tools to solve complex queries.

• **Adaptive Thinking.** We further investigate tool-use efficiency by measuring the proportion of questions where model invokes tools before and after RL. As shown in Figure 5, prior to RL, model over-relies on tools, using them for most questions. After RL, however, tool invocation rate decreases significantly, showing that model learns adaptive reasoning: it solves problems directly when tools are unnecessary while still leveraging them when beneficial. Combined with Figure 7, these results highlight that reinforcement learning improves both efficiency and flexibility, enabling the balance between textual reasoning and tool calls.

• **Training Dynamic.** We further analyze model dynamics during RL by tracking response length, reward, and tool invocation frequency throughout training (Figure 6). The average number of tool calls steadily decreases over time; however, the variance remains large, indicating that model does not simply converge to a fixed number of tool invocations (e.g., one per query). Instead, model learns adaptive thinking: it selectively invokes tools when necessary, while handling simpler problems with minimal or no tool use. For more challenging queries, the number and complexity of tool calls remain high, reflecting flexible and task-aware strategies. Shorter response lengths further indicate more efficient reasoning, allocating detailed tool-based steps only when beneficial. Together,

these findings highlight that reinforcement learning not only enhances tool-use effectiveness, while fostering diversity, complexity, and efficiency in reasoning.

## 5 CONCLUSION

In this work, we explore how to construct agentic multimodal models that can actively invoke tools and integrate them into reasoning, from the perspectives of training, dataset design, and evaluation. We introduce *DeepEyesV2* and conduct a practical two-stage training pipeline: supervised fine-tuning on a curated dataset to establish robust tool-use patterns, followed by reinforcement learning to strengthen and adapt tool invocation. Our analysis reveals task-dependent tool-use behaviors, and reinforcement learning enables more complex, context-aware tool combinations. Extensive experiments across perception, reasoning, and search benchmarks demonstrate the strong reasoning ability of *DeepEyesV2*, highlighting the advantages of combining tool invocation with reasoning.

## 6 ETHICS STATEMENT

This work aims to advance the tool-use capability of multimodal large language models (MLLMs) through reinforcement learning and carefully curated datasets. All datasets used in this study are collected from publicly available sources under appropriate licenses, and sensitive or unverifiable content is excluded during preprocessing to mitigate potential ethical risks. While our model demonstrates improved reasoning, perception, and search capabilities, it also introduces risks such as misuse for generating misleading content, over-reliance on automated reasoning, and privacy concerns when dealing with sensitive data. We emphasize that the proposed methods and released resources are strictly intended for research purposes. We strongly discourage any malicious applications, including but not limited to surveillance, misinformation, or decision-making in high-stakes scenarios without human oversight.

## 7 REPRODUCIBILITY STATEMENT

We provide detailed descriptions of our training settings, including datasets, preprocessing pipeline, and filtering strategies in Section 4.1. Hyperparameters for both SFT and RL stages are reported explicitly, covering batch size, learning rate schedules, optimizer configurations, and rollout parameters. The backbone model (Qwen2.5-VL-7B) is publicly available, and all external datasets used in our work are either publicly released or cited with appropriate references. Benchmark evaluations are conducted on established public datasets to ensure comparability.

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

# A APPENDIX

## A.1 THE USE OF LARGE LANGUAGE MODELS (LLMs)

In this work, Large Language Models (LLMs) are employed solely as auxiliary tools to refine the writing and improve the readability of the manuscript. All conceptual contributions, experimental designs, and analyses are conducted by authors. Authors bear full responsibility for the content of the paper, and affirm that the use of LLMs did not involve plagiarism, fabrication, or any form of scientific misconduct.

## A.2 TRAINING DATA

For perception-oriented tasks, we include V* (Wu & Xie, 2024), ArxivQA (Li et al., 2024c), Pixmo Counting (Deitke et al., 2024), TallyQA (Acharya et al., 2019), and SeekWorld (Tian et al., 2025), covering a wide range of scenarios such as object recognition, visual counting, and chart interpretation. For reasoning tasks, we adopt ReVisual (Chen et al., 2025b) to provide complex reasoning problems, and additionally incorporate MathCoder (Wang et al., 2023) and Retool (Feng et al., 2025) to supplement with executable code-based problem-solving examples. Besides, we also include long Chain-of-Cot (CoT) reasoning data in cold start subset. For search-related tasks, we employ MMSearch-R1 (Wu et al., 2025), which includes both image-based and text-based retrieval questions. We further include data from VGR (Wang et al., 2025c), Chain-of-Focus (Zhang et al., 2025a), and VLM-R$^3$ (Jiang et al., 2025) to strengthen the reinforcement learning corpus.

We present the distributions of our cold start and RL data in Figure 7. The cold start data is divided into four parts: perception, reasoning, search, and Long CoT, while the RL data includes perception, reasoning, and search.

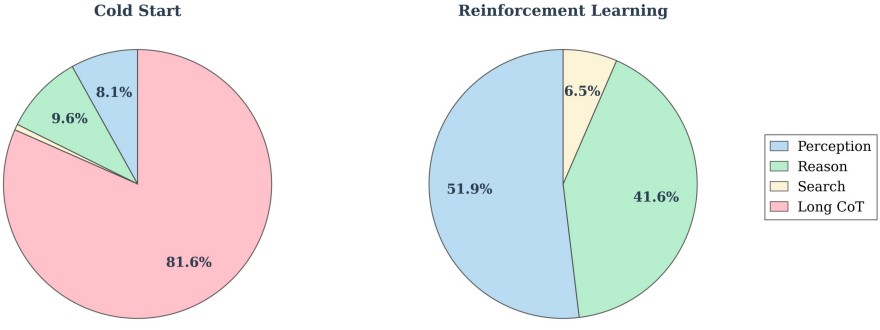

Figure 7: **Distribution of cold start and reinforcement learning data.**

## A.3 EVALUATION PROTOCOL

we summarize the benchmarks and models we compare across different kinds of tasks. Benchmarks cover three main categories: real-world understanding, mathematical reasoning, and search-intensive tasks, capturing the diversity of challenges faced by agentic multimodal models.

**Real-World & OCR & Chart Understanding.** For real-world understanding, we adopt V* (Wu & Xie, 2024), HRBench (Wang et al., 2025e), MME-RealWorld (Zhang et al., 2024b), and TreeBench (Wang et al., 2025a); for OCR, we use OCRBench (Liu et al., 2024b) and Seed-Bench-2-Plus (Li et al., 2024b); and for chart reasoning, we evaluate on CharXiv (Wang et al., 2024b) and ChartQA (Masry et al., 2022). For comparison, we include two kinds of models: (i) open-source general-purpose MLLMs, including LLaVA-OneVision (Li et al., 2024a), Qwen2.5-VL (Bai et al., 2025), and InternVL3 (Zhu et al., 2025); and (ii) grounded reasoning models, such as Pixel-Reasoner (Su et al., 2025), DeepEyes (Zheng et al., 2025b) and Thyme (Zhang et al., 2025b).

Table 6: **Results on real-world & OCR & chart understanding Benchmarks.**

| Model | Tool | Param Size | Real-World Understanding | | | | | OCR | | Chart | | |
|---|---|---|---|---|---|---|---|---|---|---|---|---|
| | | | V* Bench | HRBench 4K | HRBench 8K | MME-RealWorld | Tree Bench | OCR Bench | SEED 2 Plus | CharXiv descriptive | CharXiv reasoning | Chart QA |
| *Open-source Models* | | | | | | | | | | | | |
| LLaVA-OV | ✗ | 7B | 75.4 | 63.0 | 59.8 | 57.4 | 37.3 | - | - | - | - | 80.0 |
| Qwen2.5-VL | ✗ | 7B | 78.5 | 71.6 | 67.9 | 57.3 | 37.0 | 864 | 70.4 | 72.7 | 40.2 | 86.2 |
| Qwen2.5-VL | ✗ | 32B | 80.6 | 74.1 | 69.9 | - | 42.5 | - | **72.4** | **83.2** | 48.0 | - |
| InternVL3 | ✗ | 8B | 81.2 | 70.0 | 69.3 | - | 38.8 | 880 | 69.7 | 73.6 | 37.6 | 86.6 |
| *Proprietary Models with Tools* | | | | | | | | | | | | |
| GPT-4o | Code | - | 58.6 | 60.6 | 55.1 | - | 48.1 | 822 | 71.9 | 85.9 | 47.5 | - |
| Gemini 2.5 Pro | Code | - | 79.6 | 86.9 | - | 71.6 | 49.1 | 881 | 75.0 | 92.6 | 67.8 | - |
| *Grounded Reasoning Models* | | | | | | | | | | | | |
| Pixel-Reasoner | Crop | 7B | 84.3 | 74.0 | 66.9 | 64.4 | 39.0 | - | - | - | - | - |
| DeepEyes | Crop | 7B | **85.6** | 75.1 | 72.6 | - | 37.5 | - | - | - | - | - |
| Thyme | Code | 7B | 82.2 | 77.0 | 72.0 | 64.8 | - | 863 | - | - | - | 86.1 |
| *Agentic Multimodal Model* | | | | | | | | | | | | |
| *DeepEyesV2* | General | 7B | 81.8 | **77.9** | **73.8** | **64.9** | 42.5 | **882** | 70.5 | 78.6 | **48.9** | **88.4** |
| Δ (*vs* Qwen2.5-VL-7B) | | | +3.3 | +6.3 | +5.9 | +7.6 | +5.5 | +18 | +0.1 | +5.9 | +8.7 | +2.2 |

Table 7: **Results on multimodal reasoning benchmarks.**

| Model | Tool | Param Size | MathVista | MathVerse | MathVision | WeMath | DynaMath | LogicVista |
|---|---|---|---|---|---|---|---|---|
| *Open-source Models* | | | | | | | | |
| LLaVA-OV | ✗ | 7B | 58.6 | 19.3 | 18.3 | 20.9 | - | 33.3 |
| Qwen-2.5-VL | ✗ | 7B | 68.3 | 45.6 | 25.6 | 34.6 | 53.3 | 45.9 |
| InternVL3 | ✗ | 8B | 71.6 | 39.8 | 29.3 | 37.1 | - | 44.1 |
| *Proprietary Models with Tools* | | | | | | | | |
| GPT-4o | Code | - | 59.3 | 54.0 | - | 41.6 | 61.9 | 51.2 |
| Gemini 2.5 Pro | Code | - | 83.0 | 81.4 | - | - | - | - |
| *Text-only Reasoning Models* | | | | | | | | |
| MM-Eureka | ✗ | 7B | 72.6 | - | 28.1 | 21.8 | - | 46.3 |
| ThinkLite | ✗ | 7B | 71.6 | - | 24.6 | **41.8** | - | 42.7 |
| VL-Rethinker | ✗ | 7B | **73.7** | - | 28.4 | 36.3 | - | 42.7 |
| VLAA-Thinker | ✗ | 7B | 71.7 | - | 24.2 | 35.7 | - | 45.9 |
| *Grounded Reasoning Models* | | | | | | | | |
| DeepEyes | Crop | 7B | 70.1 | 47.3 | 26.6 | 38.9 | 55.0 | 47.7 |
| Thyme | Code | 7B | 70.0 | - | 27.6 | 39.3 | - | **49.0** |
| *Agentic Multimodal Model* | | | | | | | | |
| *DeepEyesV2* | General | 7B | 71.9 | **52.7** | **28.9** | 38.1 | **57.2** | 48.7 |
| Δ (*vs* Qwen2.5-VL-7B) | | | +3.6 | +7.1 | +3.3 | +3.5 | +3.9 | +2.8 |

**Multimodal Reasoning.** We include MathVista (Lu et al., 2023), MathVerse (Zhang et al., 2024a), MathVision (Wang et al., 2024a), WeMath (Qiao et al., 2024), and LogicVista (Xiao et al., 2024). We compare *DeepEyesV2* against existing open-source MLLMs, such as Qwen2.5-VL (Bai et al., 2025), text-only multimodal reasoning models, including MM-Eureka (Meng et al., 2025), ThinkLite (Wang et al., 2025f), VL-Rethinker (Wang et al., 2025b), and VLAA-Thinker (Chen et al., 2025a), and grounded reasoning models, such as DeepEyes (Zheng et al., 2025b) and Thyme (Zhang et al., 2025b)

**Online Searching.** We compare *DeepEyesV2* on FVQA-test (Wu et al., 2025), InfoSeek (Chen et al., 2023), MMSearch (Jiang et al., 2024), and SimpleVQA (Cheng et al., 2025). We benchmark *DeepEyesV2* against both general-purpose MLLMs such as Qwen2.5-VL (Bai et al., 2025), Gemini 2.5 Pro (Comanici et al., 2025), and GPT4o (Hurst et al., 2024), as well as models where search capability is incorporated (Jiang et al., 2024; Geng et al., 2025).

## A.4 PERFORMANCE COMPARISON WITH PROPRIETARY MODELS

In Table 6 and 7, we compare *DeepEyesV2* with existing proprietary models by using the same prompt as *DeepEyesV2*, *DeepEyesV2* achieves performance comparable to GPT-4o.

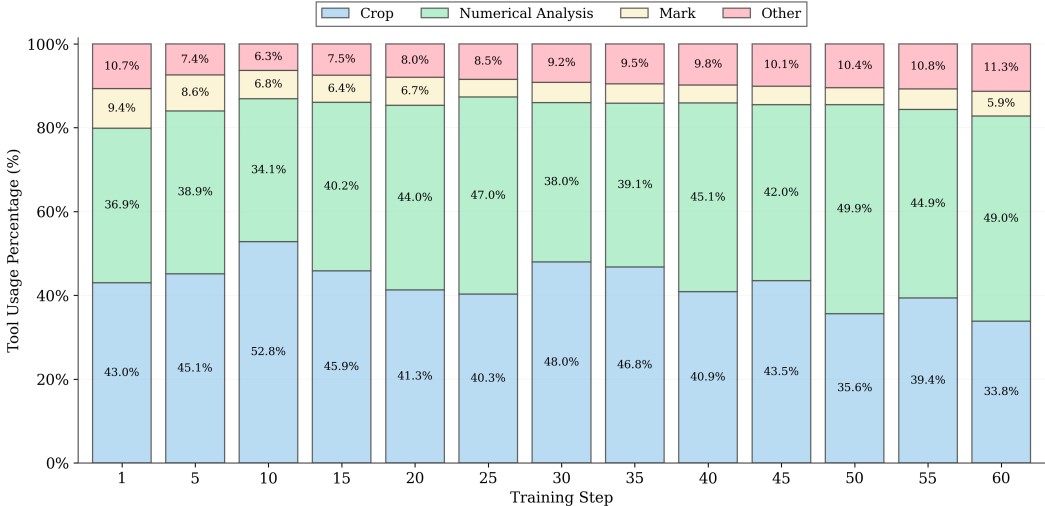

Figure 8: **Tool Usage Pattern during RL.**

| Qwen2.5-VL 7B | DeepEyes | *DeepEyesV2*-SFT | *DeepEyesV2*-RL |
|---|---|---|---|
| 16.0 | 17.3 | 18.7 | 20.8 |

Table 8: **Zero-shot results on TIR-Bench**

| *DeepEyesV2* | *DeepEyesV2* w/ Rotate |
|---|---|
| 33.5 | 38.3 |

Table 9: **Zero-shot results on OCR subset of TIR-Bench with additional tools**

## A.5 TOOL USAGE PATTERN EVOLUTION

To further understand how tool usage patterns evolve during RL training, Figure 8 presents category-wise statistics of code tools across training stages. Clear trends emerge: (i) the proportion of "Mark" tools remains largely stable, with minimal fluctuation; (ii) the proportion of "Crop" tools rises initially and then declines as training progresses; and (iii) the "Numerical analysis" and "Other" categories exhibit the opposite trajectory—decreasing at first and subsequently increasing. These trends in tool usage are broadly consistent with the changes in tool usage shown in Figure 4.

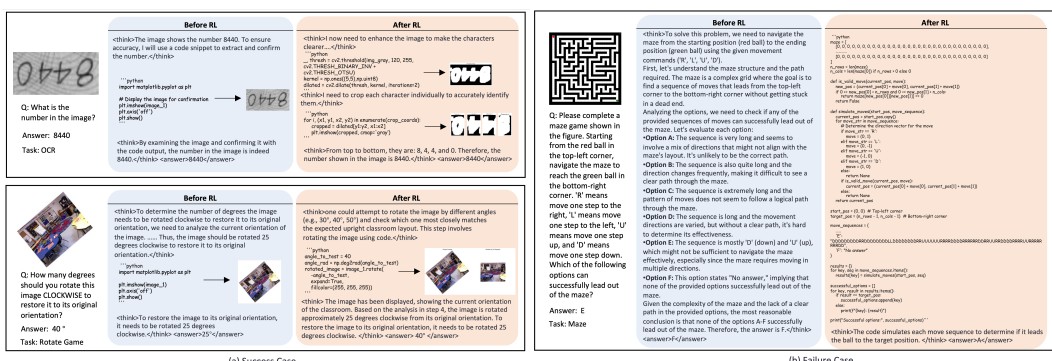

Figure 9: **Zero-shot tool utilization example of DeepEyesV2 on TIR-Bench.**

## A.6 ZERO-SHOT GENERALIZATION OF TOOL USAGE

We conduct zero-shot testing of *DeepEyesV2* on TIR-Bench Li et al. (2025a). The tasks and required tool categories in TIR-Bench are not included in our cold-start and RL training data, therefore TIR-Bench can serve as an excellent platform for zero-shot testing of *DeepEyesV2* to illustrate its generalization capability in tool invocation.

**Performance Comparison.** The performance of *DeepEyesV2* on TIR-Bench is summarized in Table 8. Specifically, *DeepEyesV2*-SFT represents the model after the cold-start stage, and *DeepEyesV2*-RL represents the model after RL training. It is evident that *DeepEyesV2* substantially outperforms the baseline (Qwen2.5-VL 7B) thanks to tool integration. This provides compelling evidence of *DeepEyesV2*'s strong generalization ability, confirming that great performance improvements persist even when applied to unseen tools and tasks.

**New Tool Visualization.** We present examples of zero-shot tool utilization from TIR-Bench in Figure 9, comparing model performance before and after RL across tasks such as OCR, image rotation, and maze solving. We omit the specific options for each question and only display the core code segments. Prior to RL, lacking exposure to relevant tasks, model merely represent the input image. Conversely, the post-RL model demonstrates robust generalization capabilities. Specifically, for OCR task, *DeepEyesV2* employs grayscaling and dilation to enhance character clarity, followed by cropping individual digits for recognition. For the rotation task, *DeepEyesV2* rotates the image to determine its original orientation angle. Notably, these tasks and tools are absent from both our cold-start and RL datasets; yet, without additional training, *DeepEyesV2* successfully comprehends and utilizes these novel tools. Furthermore, we highlight an interesting maze scenario where *DeepEyesV2* generates code to simulate pathfinding, thereby verifying each option. Although the code contains minor imperfections, we think this strongly evidences *DeepEyesV2*'s generalization potential with new tools. This comparison of tool usage pre- and post-RL effectively highlights *DeepEyesV2*'s exceptional adaptability to unseen tasks and tools.

**Function Call Generalization.** Furthermore, beyond the code generalization capabilities, *DeepEyesV2* demonstrates strong generalization in function calling. To evaluate this, we equip *DeepEyesV2* with a rotation tool on a subset of the Rotated OCR task from TIR-Bench, requiring model to utilize the tool via function calls rather than by writing code. Since images in this task are rotated, model must perform rotation prior to OCR recognition. It is worth noting that in addition to the rotation function call, *DeepEyesV2* retains the ability to invoke other tools via code generation. As shown in Table 9, providing the rotation function call leads to further performance improvements, with *DeepEyesV2* invoking the rotation tool in $15\%$ of instances. This demonstrates *DeepEyesV2*'s strong generalization capability regarding newly added tools.

### A.7  Tool Taxonomy

The tools can be categorized into three major classes:

**1. Code Execution.** Code execution covers a set of operations that require Python-based execution. We further divide it into four subtypes:

- **Crop:** extract a specific region of the input image for fine-grained analysis.

```python
cropped = image_1.crop((top, left, right, bottom))

plt.imshow(cropped)
plt.axis('off')
plt.show()
```

- **Numerical Analysis:** perform numerical computations, formula evaluation, or quantitative reasoning.

```python
import math
height = 68

w = height / math.tan(math.radians(37))

x = w / math.tan(math.radians(46))
print(f"w = {w}")
print(f"x = {x}")
```

- **Mark:** annotate or highlight regions of interest in the image to support reasoning.

```python
from PIL import ImageDraw
```

```
draw = ImageDraw.Draw(image_1)

box = (50, 50, 300, 200)
color = (255, 0, 0)
thickness = 8

draw.rectangle(box, outline=color, width=thickness)

plt.imshow(image_1)
plt.show()
```

- **Other:** other manipulation operations such as rotation, enhancement, or resizing.

```python
from PIL import ImageEnhance

enhancer = ImageEnhance.Brightness(image_1)

factor = 1.5

bright_img = enhancer.enhance(factor)

plt.imshow(bright_img)
plt.axis('off')
plt.show()
```

**2. Image Search.** Given an image query, we utilize SerpAPI to retrieve visually similar results from the web, returning candidate images with thumbnails.

**3. Text Search.** Based on a textual query, we retrieve relevant webpages and provides both titles and snippets of content.

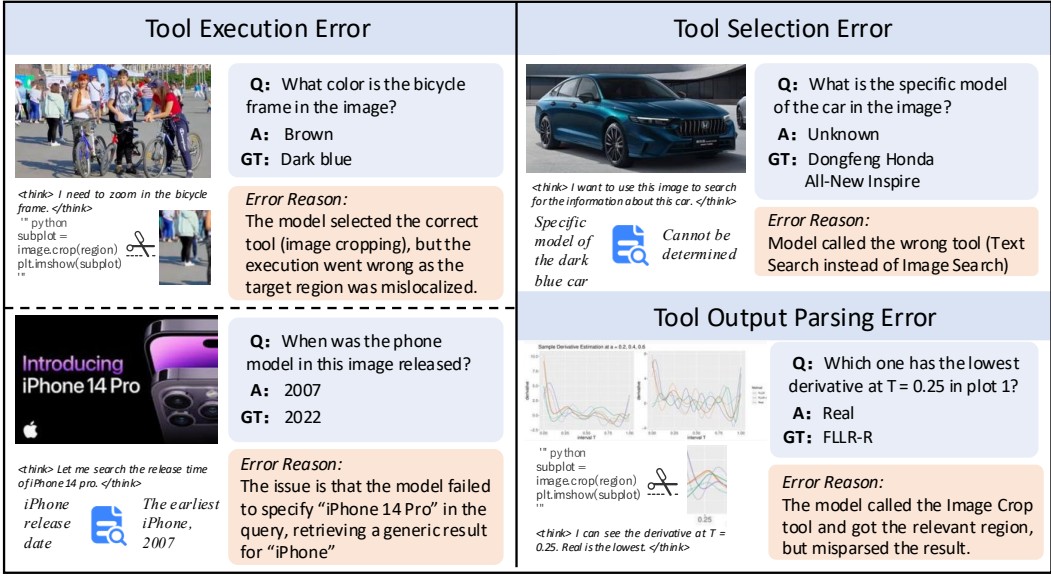

Figure 10: **Error analysis.**

## A.8 ERROR ANALYSIS

We categorize the errors made by *DeepEyesV2* into three main types (Figure 10). First, tool execution errors occur when the model generates a correct reasoning trajectory but fails during tool

operation, such as cropping the wrong region or using incorrect search keywords. Second, tool selection errors arise when the model chooses an inappropriate tool for the task, for example selecting text search when an image search is required. Third, tool result analysis errors happen when the model correctly selects and executes a tool, but misinterprets or incorrectly analyzes the returned outputs. This categorization helps to identify the main sources of failure and guides future improvements in tool-invoked reasoning.

## A.9 PROMPT

---

**SYSTEM_PROMPT**

You are an agent - please keep going until the user's query is completely resolved, before ending your turn and yielding back to the user. Only terminate your turn when you are sure that the problem is solved.

Solve the following problem step by step. In your reasoning process, if the answer cannot be determined, you can write Python code in a Jupyter Notebook to process the image and extract more information from it. The stdout and stderr content, along with the images generated by `plt.show()` will be returned to better assist with the user query.

You MUST use the python tool to analyze or transform images whenever it could improve your understanding. This includes but is not limited to zooming in, rotating, adjusting contrast, computing statistics, or isolating features.

If you find you sufficient knowledge to confidently answer the question, you MUST conduct search to thoroughly seek the internet for information. No matter how complex the query, you will not give up until you find the corresponding information.

You can conduct image search, which will trigger a Google Lens search using the original image to retrieve relevant information that can help you confirm the visual content, and text search, which will use Google Search to return relevant information based on your query.

You MUST plan extensively before each function call, and reflect extensively on the outcomes of the previous function calls. DO NOT do this entire process by making function calls only, as this can impair your ability to solve the problem and think insightfully.

Additionally, you can combine python tool with search to assist in answering questions. Python tool can help enhance your understanding of images, while search tools can provide the knowledge you lack. Please use python tool and search flexibly. However, you can only call one type of tool in a single round; you cannot use a python tool and perform a search simultaneously.

For all the provided images, in order, the i-th image has already been read into the global variable `image_i` using the `PIL.Image.open()` function. For example, the first image can be accessed as `image_1`. When writing Python code, you can directly use these variables without needing to read them again.

## Tools

## python
Your python code should be enclosed within `` `` tag.

Example for calling Python code in Jupyter Notebook:

```

```python
\# python code here
```

```

---

Note:

1. **python** can be called to analyze the image. **python** will respond with the output of the execution or time out after 300.0 seconds.

2. Like jupyter notebook, you can use Python code to process the input image and use `plt.show()` to visualize processed images in your code.

3. All python code are running in the same jupyter notebook kernel, which means the functions and variables are automatically stored after code execution.

4. You program should always returns in finite time. Do not write infinite loop in your code.

5. Writing file to disk is not allowed.

## search

You are provided with function signatures within `<tools></tools>` XML tags:

```
<tool_call>
{"type":"function", "function":
{
  "name": "image_search",
  "description": "Retrieves top 10 images and descriptions
  from Google's image search using the original image.
  Should only be used once.",
},
{
  "name": "search",
  "description": "Performs batched web searches: supply an
  array 'query'; the tool retrieves the top 10 results for
  each query in one call.",
  "parameters": {
    "type": "object",
    "properties": {
      "query": {
        "type": "string",
        "description": "Search query to find
        relevant information."
      }
    },
    "required": [
      "query"
    ]
    }
}
}
</tool_call>
```

Example for calling search: Return a json object with function name and arguments within `<tool_call></tool_call>` XML tags:

```
<tool_call>
{"name": "image_search"}
</tool_call>
<tool_call>
{"name": "search", "arguments": {"query": "Does Cloudflare
analyze submitted data to block attacks"}}
</tool_call>
```

Note:

1. You MUST engage in many interactions, delving deeply into the topic to explore all possible aspects until a satisfactory answer is found.

2. Before presenting a Final Answer, you will **cross-check** and **validate the information** you've gathered to confirm its accuracy and reliability.
3. You will carefully analyze each information source to ensure that all data is current, relevant, and from credible origins.
4. Please note that you can **only** call search once at a time. If you need to perform multiple searches, please do so in the next round.
5. You can **only** conduct image search once.

---

USER_PROMPT

{Question}
You must put your answer inside `<answer> </answer>` tags, i.e., `<answer> answer here </answer>`. Please reason step by step. Use Python code to process the image if necessary. You can conduct search to seek the Internet. Format strictly as `<think> </think>  `(if code is needed) or `<think> </think> <tool_call> </tool_call>`(if function call is needed) or `<think> <think> <answer> </answer>`.

---

RETURN_CODE_USER_PROMPT

Code execution result:

```
stdout:
```
{stdout}
```

stderr:
```
{stderr}
```

Image:
{image}
```

---

RETURN_IMAGE_SEARCH_USER_PROMPT

A Google image search for the image found 5 results:
## Web Results
1. `<image>` [{title}]
2. `<image>` [{title}]

---

RETURN_TEXT_SEARCH_USER_PROMPT

A Google search for '{query}' found 5 results:
## Web Results
1. [{title}] ({link}) {snippet}
2. [{title}] ({link}) {snippet}

