# OpenReview forum: "DeepEyesV2: Toward Agentic Multimodal Model"
_ICLR.cc/2026/Conference — ICLR 2026 Poster_

### Official Review · Reviewer_xzDS · 2025-10-26

**Soundness:** 4
**Presentation:** 3
**Contribution:** 3
**Rating:** 6
**Confidence:** 3

**Summary:**

This work presents a sound and well-executed study on building an agentic MLLM using SFT and RL, unifying code execution and web search. Through extensive experiments, the authors show that their work DeepEyesV2 presents significant gains over prior work. The analysis and ablation studies provided valuable insights into why direct RL on a base model doesn't elicit robust tool-use.

**Strengths:**

1. Extensive experiments using RL alone to post-train Qwen2.5-VL and the thorough analysis are interesting and insightful.

2. Comprehensive evaluation on 8 benchmarks and analysis. Interesting and insightful comparison between pre-RL and post-RL in terms of tool use patterns.

3. Strong results across benchmarks.

**Weaknesses:**

1. Lack of comparison with a concurrent work WebWatcher, whose core methodologies seem very similar to this work. WebWatcher's performance is cited in this work but its methodological similarities are not discussed.

2. Unclear how much of the gain on search benchmarks comes from the use of SerpAPI. Do baselines and this work use the same set of search APIs?

**Questions:**

1. Will the curated dataset be released? If not, the results would be hard to reproduce since this "carefully curated training corpus" is fundamental.

2. SerpAPI was used for image search, but it's unclear if any API was used for text search. Does it simply use Google search? If so, did you encounter rate limiting?

---

> ### Author Response · Authors · 2025-11-20
> **Response to Reviewer xzDS**
>
> Thanks for your insightful feedback and support. We sincerely appreciate your valuable comments and your recognition of the soundness of the experiments.
>
> > **Q1: Lack of comparison with a concurrent work WebWatcher, whose core methodologies seem very similar to this work. WebWatcher's performance is cited in this work but its methodological similarities are not discussed.**
>
> There is an ***essential difference*** between DeepEyesV2 and WebWatcher. DeepEyesV2 is a general agentic multimodal model, while WebWatcher is merely a search model. DeepEyesV2 has achieved leading performance on tasks such as image understanding, chart analysis, and mathematical reasoning, whereas WebWatcher has only been evaluated on search-related tasks.
>
>
> > **Q2: Unclear how much of the gain on search benchmarks comes from the use of SerpAPI. Do baselines and this work use the same set of search APIs?**
>
> In ***Table 3***, we use the same image search engine as the baseline. We summarize the comparison between our search engine and those of other works in the table below. WebWatcher uses SerpAPI for text search, which is a search engine based on Google Search and more powerful than Google Search itself. Qwen 2.5 VL Search and MMSearch-R1, after conducting searches via SerpAPI, access the content of the retrieved URLs and use an LLM to summarize the web page content, making them more powerful than SerpAPI alone. In contrast, we only use the basic Google Search for text search. Despite our text search engine being significantly weaker than those used by other models, our performance is superior. This indicates that the excellent performance does not stem from a more advanced search engine but rather from our model’s stronger reasoning capabilities. We will highlight the search engine details in the modified version.
>
> | Model | Multimodal Search | Text Search |
> | --- | --- | --- |
> | WebWatcher | SerpAPI | SerpAPI |
> | Qwen 2.5 VL Search | SerpAPI | SerpAPI + Webpage Visit + Summarizer |
> | MMSearch-R1 | SerpAPI | SerpAPI + Webpage Visit + Summarizer |
> | DeepEyesV2 | SerpAPI | Google Search |
>
>
> > **Q3: Will the curated dataset be released? If not, the results would be hard to reproduce since this "carefully curated training corpus" is fundamental.**
>
> Upon acceptance, we will release all data, training code, and pre-trained models to facilitate the reproduction of our work by the community. We appreciate your support for our work.
>
>
> > **Q4: SerpAPI was used for image search, but it's unclear if any API was used for text search. Does it simply use Google search? If so, did you encounter rate limiting?**
>
> As explained in ***our response to Q2*** above, we have described our text search engine. The specific rate limit is related to engineering implementation.

---

### Official Review · Reviewer_zneT · 2025-11-01

**Soundness:** 4
**Presentation:** 4
**Contribution:** 3
**Rating:** 4
**Confidence:** 3

**Summary:**

This paper introduces DeepEyesV2, an agentic multimodal model designed to actively invoke external tools like code execution and web search and integrate their outputs into its reasoning loop. The authors first demonstrate that direct reinforcement learning (RL) alone fails to teach robust tool use, often leading to reward hacking. To solve this, they propose a two-stage training pipeline: (1) a "cold-start" supervised fine-tuning (SFT) stage using a carefully curated dataset of difficult, tool-beneficial examples to establish reliable tool-use patterns , and (2) an RL stage to refine and adapt these skills. The experimental results showing DeepEyesV2 outperforms existing models on real-world understanding, mathematical reasoning, and search benchmarks. The analysis also reveals that the model learns task-adaptive tool invocation (e.g., image operations for perception, computation for reasoning) and that RL enables more complex, context-aware tool combinations and efficiency.

**Strengths:**

- This paper is well written and easy to follow.
- The paper goes far beyond reporting final scores by providing deep insights into the learning dynamics.
  - A discovery of "Adaptive Thinking", where the SFT model over-relies on tools, and the RL stage teaches it the efficiency of when not to use tools .
- The performance is well, the authors validate DeepEyesV2 across a wide and diverse range of benchmarks, covering real-world understanding, mathematical reasoning, and search-intensive tasks.

**Weaknesses:**

- A substantive weakness of the paper is its limited methodological novelty, as its core SFT + RL two-stage training paradigm and reasoning CoT data curation are well-established approaches for reasoning-based models, making the work feel more like a high-quality technical report than a novel research contribution.
- The paper correctly observes that the SFT model "over-relies on tools" but fails to investigate if this is merely a statistical artifact of the cold-start data's high tool-calling density. More critically, the claim that RL fosters "adaptive efficiency" by decreasing tool use is presented as a raw observation without explaining the underlying mechanism, especially since the simple reward function ($R = R_{acc} + R_{format}$) provides no explicit incentive for this emergent efficiency.
- While the paper provides strong analyses and experiments, it is questionable whether supervising only the correctness of the final answer can effectively ensure the accuracy and relevance of the tool-use process itself. For instance, if the model invokes an irrelevant or unnecessary tool but still manages to produce the correct final answer, it receives a high reward, indistinguishable from that of a model that used tools logically. This makes the reasoning process uninterpretable and unreliable.

**Questions:**

See weaknesses.

**Details Of Ethics Concerns:**

No concerns.

---

> ### Author Response · Authors · 2025-11-20
> **Response to Reviewer zneT - 1 / 3**
>
> Thanks for your insightful feedback and support. We sincerely appreciate your recognition of the experiments. We believe the following responses would resolve all concerns.  We would appreciate it very much if you could reconsider our work and support us.
>
> > **Q1: A substantive weakness of the paper is its limited methodological novelty, as its core SFT + RL two-stage training paradigm and reasoning CoT data curation are well-established approaches for reasoning-based models, making the work feel more like a high-quality technical report than a novel research contribution.**
>
> Thanks for your thoughtful comment. While two-stage training (SFT + RL) is standard in the LLM literature (such as DeepSeek and Qwen) and not claimed as our contribution, our work differs from prior art in several key ways:
>
> 1. **Code Execution for Tool Use.** Previous works such as DeepEyes and PixelReasoner expose a single tool via function calls. In contrast, DeepEyesV2 lifts this limitation by executing code, enabling flexible invocation of diverse tools. Unlike text-only works like ReTool that only are restricted to numerical computation, DeepEyesV2 also supports a range of image operations.
>
> 2. **Unified, interleavable Multi-tool Agent.** Whereas previous works can only perform image operations or search in isolation, DeepEyesV2, however, is ***the first*** to organically integrate code execution with search within a single reasoning loop, where search and code execution can interact rather than operate independently. This integration significantly raises the upper limit of reasoning. As shown in Tables 1, 2, and 3, DeepEyesV2 outperforms existing models by a considerable margin, highlighting its strong reasoning capabilities.
>
>
> 3. **Tool-benefit–driven Data Curation.** We propose a data pipeline which explicitly (a) filters by task difficulty and (b) labels whether tool use materially improves success. This benefit-oriented curation is absent in prior work.
>
> 4. **In-depth Analysis.** We analyze the dynamics of tool-use behavior in DeepEyesV2, revealing task-adaptive patterns. Besides, we also find reinforcement learning can enable more complex tool combinations and adaptive, context-aware tool invocation.
>
> We investigate how to build an agentic multimodal model from the perspective of data, training, and evaluation. The unified agent model, data methodology centered on tool-benefit, and the agentic behaviors in multimodal settings, including adaptive tool selection, cross-tool composition, and efficiency, go beyond established single-tool or single-modality extensions of the standard training template. We will highlight the novelty of our DeeyEyesV2 in the revised version.

---

> ### Author Response · Authors · 2025-11-20
> **Response to Reviewer zneT - 2 / 3**
>
> > **Q2: The paper correctly observes that the SFT model "over-relies on tools" but fails to investigate if this is merely a statistical artifact of the cold-start data's high tool-calling density. More critically, the claim that RL fosters "adaptive efficiency" by decreasing tool use is presented as a raw observation without explaining the underlying mechanism, especially since the simple reward function (R=R_acc+R_format) provides no explicit incentive for this emergent efficiency.**
>
> Thank you for this thoughtful comment. We address both parts: (i) whether the observed “over-reliance” after SFT is a statistical artifact of cold-start data, and (ii) why RL with a minimal reward leads to adaptive efficiency despite no explicit tool-cost term.
>
> 1. **Cold-start.**
>
> As we explained in ***Section 3.2***, existing models struggle to invoke tools effectively. Therefore, we need to enable models to learn various tool invocation formats and reasoning patterns through SFT. All our SFT data involves tools thus includes tool invocation examples. However, due to the varying capabilities of different models, we cannot construct customized training data for each one. Instead, we designed our SFT data to expose models to a large number of diverse tool invocation scenarios, with extensive tool-related content included. Consequently, models consistently invoke tools in every instance after SFT training. Thus, the model's over-reliance on tools after SFT is not a statistical artifact, instead, it is because the extensive tool invocation trajectories in the SFT data lead the model to consistently use tools in every instance.
>
> 2. **RL Leads to Adaptive Efficiency.**
>
> We observe that after RL training, DeepEyesV2 has developed the advanced reasoning capability of adaptive thinking. DeepEyesV2 no longer invokes tools for all questions, but instead makes adaptive choices based on the specific problem.
>
> - **Essence of RL.** Many works (such as [1, 2, 3]) interpret the core of RL as ***improving sampling efficiency***. During RL, multiple trajectories are generated for a single problem, and training seeks to increase the likelihood of specific trajectories. In our experiments, for a given problem, DeepEyesV2 produces diverse trajectories: some do not invoke tools or use them sparingly, while others rely on frequent tool invocations.
>
> - **Simple Question.** For simple questions, correct answers can be obtained without invoking tools or with only minimal tool usage. In contrast, frequent or unnecessary tool invocations may introduce additional errors, ledding to low reward. Thus, for simple questions, RL increases the probability of trajectories that involve no or minimal tool usage, which is driven by the accuracy-based reward mechanism.
>
> - **Complex Question.** In contrast, for complex questions, minimal tool usage is insufficient to generate correct answers, and only complex tool invocation trajectories can potentially lead to rewards. Therefore, model still tends to adopt multi-turn tool usage for such scenarios.
>
> This aligns with the training curves presented in ***Figure 6***: the average number of tool invocations decreases while rewards consistently increase, yet the variance in tool usage remains large. This proves that model does not collapse into a "zero-tool" state but instead learns to regulate tool invocations based on task difficulty.
>
> In summary, multiple trajectory samplings for the same question may produce both high-frequency and low-frequency tool invocation trajectories. Reinforcement learning improves the model’s sampling efficiency: for simple questions, it biases the model toward minimal tool use; for complex questions, only multiple tool invocations can ensure correct answers. Consequently, the model acquires adaptive reasoning capabilities. We will add further explanations on the efficiency improvement of RL in tool adaptability in later version based on your suggestion.

---

> ### Author Response · Authors · 2025-11-20
> **Response to Reviewer zneT - 3 / 3**
>
> > **Q3: While the paper provides strong analyses and experiments, it is questionable whether supervising only the correctness of the final answer can effectively ensure the accuracy and relevance of the tool-use process itself. For instance, if the model invokes an irrelevant or unnecessary tool but still manages to produce the correct final answer, it receives a high reward, indistinguishable from that of a model that used tools logically. This makes the reasoning process uninterpretable and unreliable.**
>
> Thank you for your insightful comment. While using only an accuracy-based reward does not guarantee the correctness of tool usage during RL process, the vast majority of tool invocations by DeepEyesV2 are meaningful.
>
>
> - **Unnecessary Tool Invocations.** We acknowledge that using only accuracy rewards may indeed lead to trajectories with unnecessary tool invocations yet correct results. This phenomenon is also prevalent in other works (such as DeepSeek R1), despite their strong reasoning capabilities, often generate trajectories with flawed reasoning processes but correct outcomes.
>
> - **Cold-start Ensures Correct Tool Use.** All tool invocations in our cold start data are meaningful and correct. Through SFT, model itself is less likely to generate meaningless tool calls.
>
>
> - **RL Implicitly Mitigates Unnecessary Tool Calls.** As stated in ***our response to Q2***, excessive tool invocations can inherently introduce errors that prevent the model from obtaining rewards, prompting it to minimize tool usage as much as possible. Fewer tool invocations, in turn, reduce the likelihood of tool-related errors. Thus, our reward mechanism implicitly mitigates unnecessary tool calls.
>
>
> - **Reward Hacking for Process Reward.** While process-based rewards on the model’s reasoning could, in principle, improve the correctness and logical consistency of tool usage, they introduce significant practical challenges and risks of reward hacking. A rule-based evaluator would require an unwieldy set of hand-crafted rules, making development and maintenance impractical. Using an LLM to score reasoning would demand additional preference data to train a reliable judge model and still leaves room for reward hacking. Additionally, similar conclusions are presented in DeepSeek-R1: it is challenging to evaluate and supervise the accuracy of the reasoning process. Given these trade-offs, we rely solely on an accuracy-based reward.
>
> In summary, while using only accuracy as the reward for RL may potentially lead to unnecessary tool invocations, our cold start data ensures the model avoids meaningless tool calls to a certain extent, and our RL framework can implicitly suppress such invocations. We will add explanations regarding the accuracy of tool usage in revised versions.
>
>
>
> [1] Does Reinforcement Learning Really Incentivize Reasoning Capacity in LLMs Beyond the Base Model? NeurIPS 2025
> [2] The Illusion of Thinking: Understanding the Strengths and Limitations of Reasoning Models via the Lens of Problem Complexity
> [3] SFT Memorizes, RL Generalizes: A Comparative Study of Foundation Model Post-training

---

### Official Review · Reviewer_UjKn · 2025-11-01

**Soundness:** 2
**Presentation:** 3
**Contribution:** 2
**Rating:** 4
**Confidence:** 3

**Summary:**

This paper introduces DeepEyesV2, an agentic multimodal model that extends beyond traditional perception and reasoning capabilities to actively invoke external tools such as code execution environments and web search APIs. The authors observe that direct reinforcement learning alone fails to induce robust tool-use behavior, motivating a two-stage training pipeline: (1) a cold-start stage using supervised fine-tuning to establish initial tool-use patterns, and (2) a reinforcement learning stage to refine tool invocation strategies. The paper curates a diverse, moderately challenging training dataset with examples where tool use is beneficial, and validates DeepEyesV2 across multiple benchmarks including visual understanding, mathematical reasoning, and search-intensive tasks. A key finding is that DeepEyesV2 exhibits task-adaptive tool invocation, selectively using image operations for perception tasks and numerical computations for reasoning tasks, with reinforcement learning enabling complex tool combinations.

**Strengths:**

1. Good presentaion demonstrates task-adaptive tool invocation (image ops for perception, computation for reasoning)
Practical two-stage training approach (cold-start SFT + RL refinement)
2. Task-Adaptive Behavior: The finding that models learn to selectively invoke different tools based on task requirements (image operations for perception, computation for reasoning) is interesting and suggests genuine understanding rather than blind tool use.

**Weaknesses:**

1. Insufficient analysis of what RL learns: The paper lacks detailed examination of how tool-use patterns evolve during training, what new behaviors emerge, and when the model makes mistakes in tool invocation. Learning curves and failure analysis would strengthen the claims.
2. Inadequate efficiency analysis: Missing quantitative data on tool call frequency, success rates, and computational overhead.
3. Unclear generalization to novel tools: Evaluation limited to a fixed tool set. Whether the model can adapt to new tools or requires retraining remains unexplored.
4. Limited novelty in training paradigm: The two-stage SFT + RL approach is standard in LLM literature. The paper needs clearer articulation of what makes multimodal tool use fundamentally different or novel techniques beyond established methods.

**Questions:**

1. Can you provide ablation studies showing performance gains from RL versus SFT only, and analyze the primary failure modes in tool invocation?
2. What are the quantitative metrics on tool call frequency, success rates, and computational overhead (training time and inference latency)?
3. Can the model generalize to new tools not seen during training, and if so, how much additional data is required?
4. What specific reward design prevents reward hacking, and what makes multimodal tool use fundamentally different from text-only approaches?

---

> ### Author Response · Authors · 2025-11-20
> **Response to Reviewer UjKn - 1 / 3**
>
> We sincerely appreciate your recognition of the analysis of our work. We believe the following responses would resolve all concerns. We would appreciate it very much if you could reconsider our work and support us.
>
>
>
>
> > **Q1: Insufficient analysis of what RL learns: The paper lacks detailed examination of how tool-use patterns evolve during training, what new behaviors emerge, and when the model makes mistakes in tool invocation. Learning curves and failure analysis would strengthen the claims.**
>
> For a detailed investigation of **tool errors**, we summarize representative cases and provide examples in ***Appendix Figure 9***. Additionally, we have add an analysis of changes in the model's **tool usage patterns** during RL training in ***Appendix Figure 8***, based on the counts of tool types invoked in code. The statistics reveal clear trends in the proportions of different tool categories over the course of training:
> - The proportion of "Mark" tools remains relatively stable, with no significant fluctuations.
> - The proportion of "Crop" tools first increases and then decreases as the training progresses.
> - In contrast, the “Numerical analysis” and “Other” categories exhibit the opposite pattern—initially decreasing and then increasing.
>
> These trends are broadly consistent with the changes in tool invocation proportions before and after RL training that we presented in ***Figure 4***.
>
> Another notable phenomenon during RL training is the steady increase in the model’s tendency to invoke new tools, spanning diverse behaviors. We observe several interesting types, such as enhancement (e.g., adjusting image brightness or contrast), API access (e.g., retrieving stock information via code as shown in ***Figure 1***), and edge detection (e.g., using edge detection operators from OpenCV).
>
> We will further summarize the new tool-invocation behaviors and add content about these new tools in the revised manuscript.
>
>
> > **Q2: Inadequate efficiency analysis: Missing quantitative data on tool call frequency, success rates, and computational overhead.**
>
> 1. **Tool Frequency.** We present quantitative metrics comparing tool invocation frequency before and after RL training in ***Figure 5***.
>
> 2. **Tool Success Rate.** Defined as the proportion of tool invocations without errors, this rate is approximately 95% on evaluation set.
>
> 3. **Computational Overhead.** We calculated the proportion of tool-related tokens in the model’s total output tokens, which is around 18% (i.e., 18% of the output tokens correspond to tool invocations).
>
> 4. **Deployment Configuration.** The tool server is deployed on a machine equipped with 16 CPU cores.
>
> 5. **Training Time.** Using 32 H100 GPUs, each training step takes approximately 1 hour.
>
> We will add the efficiency analysis in the later version.
>
>
> > **Q3: Unclear generalization to novel tools: Evaluation limited to a fixed tool set. Whether the model can adapt to new tools or requires retraining remains unexplored.**
>
> In fact, DeepEyesV2 does not invoke tools via function calls, but through ***code execution***. This code-based approach endows DeepEyesV2 with strong tool generalization capabilities, enabling it to generate numerous tools that does not appear in cold-start data.
>
>
> 1. **Diverse Tool Category.** Tools are implemented through code execution, which inherently enables diverse tool functionality and extensibility.
>
> 2. **Comprehensive Evaluation Breadth.** Our evaluation is not constrained to a fixed, pre-defined tool set. We assess DeepEyesV2 across 4 distinct task categories spanning 20 benchmarks, ensuring coverage of a broad spectrum of tools rather than a narrow, predetermined subset.
>
> 3. **Tool Usage Distribution.** ***Figure 4*** illustrates the tool invocation distribution, where the "other" category encompasses various operations including rotation, image enhancement, and additional functionalities. This demonstrates DeepEyesV2's capability to leverage a wide range of tool types in practice.
>
> 4. **Emergent Tool Capabilities.** ***Figure 1*** showcases an example of tool generalization: DeepEyesV2 autonomously generates code to query a stock-price API, a capability not present in the cold-start training data but acquired through reinforcement learning. This illustrates that DeepEyesV2 can develop new tool usage patterns without extra cold-start data.
>
> 5. **Extended Tool Case Studies.** We will include additional case studies demonstrating novel tool applications to further substantiate DeepEyesV2's tool generalization capabilities.
>
> In summary, DeepEyesV2 exhibits strong tool generalization ability enabled by code execution. Our comprehensive experiments and case studies demonstrate zero-shot adoption of previously unseen tools. We will emphasize this distinctive tool generalization capability more prominently in the revised manuscript to better highlight the key advantages of DeepEyesV2.

---

> ### Author Response · Authors · 2025-11-20
> **Response to Reviewer UjKn - 2 / 3**
>
> > **Q4: Limited novelty in training paradigm: The two-stage SFT + RL approach is standard in LLM literature. The paper needs clearer articulation of what makes multimodal tool use fundamentally different or novel techniques beyond established methods.**
>
> Thanks for your thoughtful comment. While two-stage training (SFT + RL) is standard in the LLM literature (such as DeepSeek and Qwen) and not claimed as our contribution, our work differs from prior art in several key ways:
>
> 1. **Code Execution for Tool Use.** Previous works such as DeepEyes and PixelReasoner expose a single tool via function calls. In contrast, DeepEyesV2 lifts this limitation by executing code, enabling flexible invocation of diverse tools. Unlike text-only works like ReTool that only are restricted to numerical computation, DeepEyesV2 also supports a range of image operations.
>
> 2. **Unified, interleavable Multi-tool Agent.** Whereas previous works can only perform image operations or search in isolation, DeepEyesV2, however, is ***the first*** to organically integrate code execution with search within a single reasoning loop, where search and code execution can interact rather than operate independently. This integration significantly raises the upper limit of reasoning. As shown in Tables 1, 2, and 3, DeepEyesV2 outperforms existing models by a considerable margin, highlighting its strong reasoning capabilities.
>
>
> 3. **Tool-benefit–driven Data Curation.** We propose a data pipeline which explicitly (a) filters by task difficulty and (b) labels whether tool use materially improves success. This benefit-oriented curation is absent in prior work.
>
> 4. **In-depth Analysis.** We analyze the dynamics of tool-use behavior in DeepEyesV2, revealing task-adaptive patterns. Besides, we also find reinforcement learning can enable more complex tool combinations and adaptive, context-aware tool invocation.
>
> We investigate how to build an agentic multimodal model from the perspective of data, training, and evaluation. The unified agent model, data methodology centered on tool-benefit, and the agentic behaviors in multimodal settings, including adaptive tool selection, cross-tool composition, and efficiency, go beyond established single-tool or single-modality extensions of the standard training template. We will highlight the novelty of our DeeyEyesV2 in the revised version.
>
>
>
>
>
> > **Q5: Can you provide ablation studies showing performance gains from RL versus SFT only, and analyze the primary failure modes in tool invocation?**
>
> ***Table 5*** presents an ablation study comparing the cold-start model (DeepEyesV2-SFT) with the final model after RL. RL delivers substantial performance improvements. We also document the primary failure modes in ***Appendix Figure 9***.
>
>
> > **Q6: What are the quantitative metrics on tool call frequency, success rates, and computational overhead (training time and inference latency)?**
>
> 1. **Tool call frequency** is defined as follows: if the model invokes a tool at least once in a single trajectory, it is marked as having used the tool.
>
> 2. **Tool success rate** refers to the proportion of successful tool executions, indicating whether the code runs without errors or if the API call succeeds. This rate is approximately 95% on evaluation set.
>
> 3. The **training time** corresponds to the duration of one step in RL. Each step takes approximately 1 hour when using 32 H100 GPUs.
>
> 4. **Inference latency** refers to the proportion of tool-invocation tokens in the total number of tokens. Approximately 18% of the tokens output by DeepEyesV2 are tool-invocation tokens.
>
> 5. **Deployment Configuration.** Regarding the time required for tool invocation, each call returns results within an average of 5 seconds. We deployed our tool-invocation server on a machine equipped with 16 CPU cores.
>
> > **Q7: Can the model generalize to new tools not seen during training, and if so, how much additional data is required?**
>
> As mentioned in ***the response to Q3***, DeepEyesV2 has strong tool generalization ability and can support various types of tools through code execution. Moreover, reinforcement learning enables the emergence of tool types that do not exist in cold start scenarios. Thus, DeepEyesV2 can directly generalize to unseen tools during training without requiring any additional SFT data.

---

> ### Author Response · Authors · 2025-11-20
> **Response to Reviewer UjKn - 3 / 3**
>
> > **Q8: What specific reward design prevents reward hacking**
>
> In our final RL experiment, we only use the simplest accuracy reward and format reward, which can largely avoid reward hacking. Under our setup, we employed an LLM for verification. Reward hacking only occurs when the verification error rate is relatively high. However, since all our training data consists of QA data (which is easy to verify) and our verification achieves an accuracy of over 95%, reward hacking is highly unlikely to happen.
>
>
>
>
> > **Q9: what makes multimodal tool use fundamentally different from text-only approaches?**
>
> Multimodal tools are fundamentally different from text-only approaches. Multimodal tools are an extension of multimodal thinking: they compensate for the model’s native perceptual limits and expand its reasoning substrate from text into pixels. Text tools primarily serve two roles: information acquisition (web search/RAG) and execution over text. However, multimodal tools enable model to use images to think: to act on pixel space (crop, mark, enhance), extract fine-grained visual evidence (read axes, locate points/objects, measure distances/areas), and carry out numerical computations over those measurements within the same reasoning loop.
>
> This distinction matters in practice:
>
> - **Multimodal tool use is not just extra retrieval;** it is pixel-grounded analysis. Agent can iteratively interleave visual operations, numeric reasoning, and image/text search, feeding tool outputs back into the chain of thought, which can not realized by text-only pipelines fundamentally.
>
> - **It bridges perception and reasoning.** Visual manipulation and measurement turn perception into a computational step, making quantitative visual reasoning (e.g., chart/OCR/math problems) tractable rather than function call.
>
> - **It changes the decision landscape.** Agent must decide if tools are needed, which visual parameters to use (where to crop/mark), whether to search by image or text, and how to fuse visual artifacts with retrieved knowledge—introducing failure modes and learning challenges unique to multimodality.
>
>
> In short, multimodal tools extend the model’s reasoning medium from text to pixels. They are designed to think with images: acquiring, transforming, and quantifying visual evidence, thereby addressing recognition limits and enabling reliable, interpretable multimodal reasoning that text-only approaches intrinsically cannot provide.

---

> > ### Comment · Reviewer_UjKn · 2025-11-27
> >
> > Thank you for the careful reply and for taking the time to explain the method in more detail. I appreciate the effort in addressing the comments.
> >
> > I still have concerns about the adaptive behavior and the generalization ability of the method.
> >  After reading the paper again and reading the reply, I still do not see clear proof that DeepEyesV2 can use new tools in a real zero-shot way at inference time.
> >
> > 1. The reply gives some examples, such as the stock-price API code in Figure 1, but this is only one case and not a controlled test. There is no experiment where the model is given a truly new tool, such as OCR or another unseen operation, and must use it correctly without extra training.
> > 2. The paper states that code execution “inherently generalizes,” but this does not show real generalization. Code execution only means the model can output code. It does not show that the model can understand a new tool, pick the right arguments, and use it correctly in a new task.
> > 3. RL changes tool frequency (Figure 5) and reduces tool use on easy questions. But this only reflects behavior within the fixed tool set and does not show adaptivity for unseen tools.
> >
> > Because of this, the claims about “adaptive behavior” and “generalization” are not yet supported by strong evidence.  Adding more tools and allowing code execution does not itself prove strong novelty.
> >  Without real zero-shot tests or clear evidence of new adaptive behavior, the key contributions remain uncertain. This is still my main concern. Based on this, I will keep my score for now.

---

> ### Author Response · Authors · 2025-11-28
> **Response to Reviewer UjKn's Concern on Generalization 1 / 2**
>
> We appreciate your constructive comments. To demonstrate the strong generalization capability of DeepEyesV2 regarding tool execution, we conduct experiments on TIRBench [1]. TIRBench is a benchmark designed to comprehensively evaluate model's 'think-with-image' capabilities, encompassing tasks such as maze solving and rotation. Crucially, neither the tasks nor the tool categories present in TIRBench are included in our cold-start or RL training datasets. Therefore, we consider TIRBench to be an excellent benchmark for conducting a **zero-shot** evaluation of DeepEyesV2.
>
> 1. **Performance Comparison.**
>
> The performance of DeepEyesV2 on TIRBench is summarized in the following table. Specifically, DeepEyesV2 SFT represents the model after the cold-start stage, and DeepEyesV2 RL represents the model after RL training. It is evident that DeepEyesV2 substantially outperforms the baseline (Qwen2.5VL 7B) thanks to tool integration. This provides compelling evidence of DeepEyesV2's strong generalization ability, confirming that great performance improvements persist even when applied to **unseen tools and tasks**.
>
> | Model | Acc |
> | --- | --- |
> | Qwen2.5VL 7B | 16.0 |
> | DeepEyes | 17.3 |
> | DeepEyesV2 SFT | 18.7 |
> | DeepEyesV2 RL | 20.8 |
>
>
> 2. **New Tool Visualization.**
>
> We present examples of zero-shot tool utilization from TIRBench in Figure [1](https://imgur.com/ZNWjzmh
> ) and [2](https://imgur.com/CPsDZXW), comparing model performance before and after RL across tasks such as OCR, image rotation, and maze solving (You can click the numbers to navigate.). We omit the specific options for each question and only display the core code segments. Prior to RL, lacking exposure to relevant tasks, model merely represent the input image. Conversely, the post-RL model demonstrates robust generalization capabilities. Specifically, for OCR task, DeepEyesV2 employs grayscaling and dilation to enhance character clarity, followed by cropping individual digits for recognition. For the rotation task, DeepEyesV2 rotates the image to determine its original orientation angle. Notably, these tasks and tools are absent from both our cold-start and RL datasets; yet, without additional training, DeepEyesV2 successfully comprehends and utilizes these novel tools. Furthermore, we highlight an interesting maze scenario where DeepEyesV2 generates code to simulate pathfinding, thereby verifying each option. Although the code contains minor imperfections, we think this strongly evidences DeepEyesV2's generalization potential with new tools. This comparison of tool usage pre- and post-RL effectively highlights DeepEyesV2's **exceptional adaptability to unseen tasks and tools**.
>
> 3. **Function Call Generalization.**
>
> Furthermore, beyond the code generalization capabilities, DeepEyesV2 demonstrates **strong generalization in function calling**. To evaluate this, we equip DeepEyesV2 with a rotation tool on a subset of the Rotated OCR task from TIRBench, requiring model to utilize the tool via function calls rather than by writing code. Since images in this task are rotated, model must perform rotation prior to OCR recognition. It is worth noting that in addition to the rotation function call, DeepEyesV2 retains the ability to invoke other tools via code generation. As shown in the table below, providing the rotation function call leads to further performance improvements, with DeepEyesV2 invoking the rotation tool in **15%** of instances. This demonstrates DeepEyesV2's strong generalization capability regarding newly added tools.
>
>
>
> | Model | Acc |
> | --- | --- |
> | DeepEyesV2 | 33.5 |
> | DeepEyesV2 w/ Rotate | 38.3 |
>
>
>
>
> We conducted **zero-shot testing** on a completely unseen benchmark to compare tool invocation patterns before and after RL. Complemented by additional function call experiments, these results fully demonstrate DeepEyesV2's **powerful tool generalization capabilities, encompassing not only the understanding of unseen tools but also proficiency in function calling.**
>
> [1] TIR-Bench: A Comprehensive Benchmark for Agentic Thinking-with-Images Reasoning

---

> > ### Author Response · Authors · 2025-11-28
> > **Response to Reviewer UjKn's Concern on Generalization 2 / 2**
> >
> > We provide further elaboration on DeepEyesV2's tool generalization capabilities. Specifically, we categorize this generalization into three types:
> >
> >
> > 1. **Fixed System Prompt and Novel Tool.**
> >
> > First, using a fixed system prompt, DeepEyesV2 can **directly and correctly invoke entirely new tools** to solve unseen tasks. This capability is fully demonstrated by the performance comparison table and the figure illustrating tool invocation examples presented above.
> >
> >
> >
> > 2. **Generalization to New Function Call.**
> >
> > Second, regarding additional new function-based tools, we modify the system prompt to introduce these tools. DeepEyesV2 is able to correctly invoke them, resulting in performance improvements. This demonstrates that DeepEyesV2 also possesses **strong generalization capabilities for tools presented in a function format**.
> >
> >
> > 3. **Data Efficiency.**
> >
> > Finally, we address generalization regarding training data. Unlike the original DeepEyes, which was restricted to invoking a single tool, DeepEyesV2 employs code to call a diverse range of tools. Yet, DeepEyesV2 is trained on a dataset significantly smaller than that of DeepEyes. We think this data efficiency further underscores DeepEyesV2's robust generalization capabilities.
> >
> >
> > In summary, we think DeepEyesV2 demonstrates **strong tool generalization capabilities**, allowing it to correctly invoke tools even for unseen task types and novel tools. We hope the additional experiments and case studies effectively address your concerns. We would be extremely grateful if you could reconsider our work and support our paper.

---

### Official Review · Reviewer_VFi3 · 2025-11-03

**Soundness:** 3
**Presentation:** 3
**Contribution:** 3
**Rating:** 8
**Confidence:** 3

**Summary:**

The paper introduces DeepEyesV2, an agentic multimodal model designed to actively use external tools—such as code execution and web search—within its reasoning process. Unlike conventional multimodal large language models that passively interpret visual and textual inputs, DeepEyesV2 integrates tool invocation into a closed reasoning loop.

**Strengths:**

- The work clearly articulates the challenge of building agentic multimodal models and distinguishes itself from prior “thinking-with-image” paradigms by integrating multiple heterogeneous tools (code + search) in a unified reasoning loop.
- The proposed two-stage pipeline (cold-start + RL) is well-motivated and empirically justified, addressing the instability of direct RL for tool learning.
- DeepEyesV2 consistently outperforms both general-purpose and tool-augmented baselines, often matching or exceeding larger models in accuracy.
- The behavioral study of tool distribution and adaptive invocation provides convincing evidence that the model learns non-trivial reasoning strategies rather than merely following scripted patterns.

**Weaknesses:**

- The success of the approach relies heavily on the curated cold-start dataset. Details about data sources, annotation quality, and scalability are somewhat underexplored.
- Most benchmarks are vision-centric. It remains unclear how well the approach generalizes to other modalities such as audio, video, or 3D tasks.

**Questions:**

- How robust is DeepEyesV2’s tool invocation under noisy or adversarial tool outputs (e.g., failed code execution or irrelevant search results)?
- How does DeepEyesV2 compare to closed-source agentic systems like GPT-4o with “thinking with images” capabilities under controlled conditions?
- Is there any mechanism to control when not to invoke a tool to reduce unnecessary computational overhead?

---

> ### Author Response · Authors · 2025-11-20
> **Response to Reviewer VFi3 - 1 / 2**
>
> We sincerely appreciate your positive comments on our motivation, method, and experiments, along with your thoughtful feedback and suggestions. We hope our response can address your concerns.
>
>
> > **Q1: The success of the approach relies heavily on the curated cold-start dataset. Details about data sources, annotation quality, and scalability are somewhat underexplored.**
>
> 1. **Data Sources.** We have enumerated all sources by category in ***Appendix A.2***.
>
> 2. **Annotation Quality.** We apply a multi-stage filtering pipeline to ensure trajectory quality. (1) **Verified GT.** We retain only prompts with correct ground truth. For each prompt, GPT-4o generates an answer, and we keep the prompt only if the GPT-4o answer matches the ground truth. (2) **Correct Trajectory Answer.** We keep only trajectories whose final result is are consistent with the ground truth. (3) **Correct and verified Tool Invocation.** Trajectories must include explicit tool calls (e.g., code execution or search). Every tool call is executed in a sandbox and must complete without runtime errors. Samples with invalid tool usage (e.g., placeholder code or incorrect API calls) are discarded. Through automatic correctness filtering, we preserve trajectories that are both correct and contain valid tool calls, ensuring the quality of the cold-start data.
>
> 3. **Scalability.** The entire pipeline, from data filtering and trajectory generation to final validation, is fully automated and requires no manual intervention, making it easy to scale.
>
> Thanks for your advice. We will emphasize all details of our cold start data in the revised paper.
>
>
> > **Q2: Most benchmarks are vision-centric. It remains unclear how well the approach generalizes to other modalities such as audio, video, or 3D tasks.**
>
> 1. **Easy Generalize to Other Modalities.**
> DeepEyesV2’s code execution tool is inherently generalizable: code can be used to process not only images but also video, audio, and 3D data. However, compared with fixed function calls, coding comes from the models' native capability, which is supported by massive amount of pretrain and intruction-finetuning data. Code-based tools offer greater flexibility and stronger scalability. Therefore, in the form of code, DeepEyesV2 can be easily extended to other modalities.
>
>
> 2. **Limitation of Base Model.**
> The base model we use (Qwen2.5-VL) can not handle audio or 3D inputs. Recent works, such as GRIT (NeurIPs 2025), also mainly focus on vision-centric tasks. While it is a common practice to adopt Qwen2.5-VL as base model, we believe it is a promising direction to switch to other models such as Qwen Omni, which supports additional modalities, and extend our evaluation to audio, video, and 3D tasks.
>
>
>
>
>
>
> > **Q3: How robust is DeepEyesV2’s tool invocation under noisy or adversarial tool outputs (e.g., failed code execution or irrelevant search results)?**
>
> We observe two types of model behavior when facing noicy environmental feedback:
>
>
> 1. **Adjusting Tool Parameters.** For example, if generated code raises an error, DeepEyesV2 attempts to fix the bug and rerun
>
> 2. **Switching Tools.** For example, if a code-based search fails, DeepEyesV2 falls back to a direct search tool
>
> We observed that 5% of the data contains has error feedback from the environment, either caused by model generated erroneous code or execution timeout. Among all these data, for 80% of tool-invocation failures, DeepEyesV2 re-invokes tools to correct the error, underscoring its robustness against noisy input.
>
> We attribute DeepEyesV2’s robustness to noisy feedback to two main factors:
>
> 1. The SFT dataset exposes model to noisy-feedback scenarios, enabling it to learn stable behavior under imperfect supervision.
> 2. During RL training, noisy feedback naturally arises for various reasons, and model learns to identify, discount, and correct such errors through iterative updates.
>
> We will add the analysis of robustness in the revised version.
>
>
>
>
>
> > **Q4: How does DeepEyesV2 compare to closed-source agentic systems like GPT-4o with “thinking with images” capabilities under controlled conditions?**
>
>
> We have added comparisons with existing closed-source models in **Appendix Section A.4***, and partial results are presented in the table below. Using the same prompts as DeepEyesV2, we can observe that the performance of DeepEyesV2 is essentially on par with that of GPT-4o.
>
> | Benchmark | GPT4o w/ Tools | DeepEyesV2 |
> | --- | --- | --- |
> | HRBench4K | 60.6 | 77.9 |
> | SEED2 Plus | 71.9 | 70.5 |
> | Charxiv reason | 47.5 | 48.9 |
> | MathVerse | 54.0 | 52.7 |
> | WeMath | 41.6 | 38.1 |

---

> ### Author Response · Authors · 2025-11-20
> **Response to Reviewer VFi3 - 2 / 2**
>
> > **Q5: Is there any mechanism to control when not to invoke a tool to reduce unnecessary computational overhead?**
>
> whether to invoke a tool is entirely decided by the model itself; we do not impose hard constraints, thresholds, or call budgets. Forcing strict control over tool calls can degrade accuracy on multi-step reasoning and complex visual understanding tasks, so we keep the default unconstrained for the main results.
>
> We will refer to the budget control methods in GPT-OSS and SEED to study how to control model tool calls. Thanks for your suggestions.

---

### Author Response · Authors · 2025-11-20
**General Response - 1 / 3**

We thank the AC for organizing the review process and all reviewers for their time and thoughtful evaluations. We have also summarized the strengths of our work to highlight the advantages.

> **Strengths**
1. **Novelty.** ***Reviewer VFi3*** acknowledged the novelty of DeepEyesV2.
2. **Performance.** ***All reviewers*** acknowledged the strong performance of DeepEyesV2.
3. **Sufficient In-depth Analysis.** ***All reviewers*** considered our analysis to be thorough and in-depth.

We greatly appreciate reviewers' recognition of the novelty, strong performance, and thorough analysis of our work. Below, we will provide responses to some general questions to further emphasize the contributions of DeepEyesV2 and offer deeper analyses. Besides, we also add the **analysis of RL** and **performance comparison with proprietary models** in ***Appendix*** in **blue**.

> **Novelty**

We adopt a two-stage training approach combining SFT and RL, which is standard in LLM literature (such as DeepSeek and Qwen), but we do not claim this training method as our innovation. Our core novelty and key contribution is being the first to develop a multimodal agent model that organically integrates code execution and search. We summarize our novelty as follows:

1. **Code Execution for Tool Use.** Previous works such as DeepEyes and PixelReasoner expose a single tool via function calls. In contrast, DeepEyesV2 lifts this limitation by executing code, enabling flexible invocation of diverse tools. Unlike text-only works like ReTool that only are restricted to numerical computation, DeepEyesV2 also supports a range of image operations.

2. **Unified, interleavable Multi-tool Agent.** Whereas previous works can only perform image operations or search in isolation, DeepEyesV2, however, is ***the first*** to organically integrate code execution with search within a single reasoning loop, where search and code execution can interact rather than operate independently. This integration significantly raises the upper limit of reasoning. As shown in Tables 1, 2, and 3, DeepEyesV2 outperforms existing models by a considerable margin, highlighting its strong reasoning capabilities.

3. **Tool-benefit–driven Data Curation.** We propose a data pipeline which explicitly (a) filters by task difficulty and (b) labels whether tool use materially improves success. This benefit-oriented curation is absent in prior work.

4. **In-depth Analysis.** We analyze the dynamics of tool-use behavior in DeepEyesV2, revealing task-adaptive patterns. Besides, we also find reinforcement learning can enable more complex tool combinations and adaptive, context-aware tool invocation.

We investigate how to build an agentic multimodal model from the perspective of data, training, and evaluation. The unified agent model, data methodology centered on tool-benefit, and the agentic behaviors in multimodal settings, including adaptive tool selection, cross-tool composition, and efficiency, go beyond established single-tool or single-modality extensions of the standard training template.





> **Generalization of Tool Use**

Since we utilize ***code*** instead of function call for tool execution, our tool invocation offers strong generalization and does not require specific cold-start data.

1. **Diverse Tool Category.** Tools are implemented through code execution, which inherently enables diverse tool functionality and extensibility.

2. **Comprehensive Evaluation Breadth.** Our evaluation is not constrained to a fixed, pre-defined tool set. We assess DeepEyesV2 across 4 distinct task categories spanning 20 benchmarks, ensuring coverage of a broad spectrum of tools rather than a narrow, predetermined subset.

3. **Tool Usage Distribution.** ***Figure 4*** illustrates the tool invocation distribution, where the "other" category encompasses various operations including rotation, image enhancement, and additional functionalities. This demonstrates DeepEyesV2's capability to leverage a wide range of tool types in practice.

4. **Emergent Tool Capabilities.** ***Figure 1*** showcases an example of tool generalization: DeepEyesV2 autonomously generates code to query a stock-price API, a capability not present in the cold-start training data but acquired through reinforcement learning. This illustrates that DeepEyesV2 can develop new tool usage patterns without extra cold-start data.

5. **No Need for Additional Data.** DeepEyesV2 has strong tool generalization ability and can support various types of tools through code execution. Moreover, reinforcement learning enables the emergence of tool types that do not exist in cold start scenarios. Thus, DeepEyesV2 can directly generalize to unseen tools during training without requiring any additional SFT data.

In summary, DeepEyesV2 exhibits strong tool generalization ability enabled by code execution. Our comprehensive experiments and case studies demonstrate zero-shot adoption of previously unseen tools.

---

> ### Author Response · Authors · 2025-11-20
> **General Response - 2 / 3**
>
> > **Analysis of RL**
>
> DeepEyesV2 exhibits a variety of interesting behaviors during RL process, which has significantly advanced the intelligence ceiling of the model's reasoning. We conduct further analysis of the RL process to provide explanations for the interesting behaviors exhibited by DeepEyesV2 during RL training.
>
>
> 1. **New Tool Behaviour.**
>
> We present the evolution of model’s tool-invocation patterns during RL training in ***Appendix Figure 8***, which reveals clear trends:
>
> - The invocation frequency of the “Crop” tool first increases and then decreases.
> - The “Numerical analysis” and “Other” tools show the opposite pattern: they decrease initially and then increase.
>
> These trends align with the changes in tool-usage proportions before and after RL training, as shown in ***Figure 4***. We observe several interesting types, such as enhancement (e.g., adjusting image brightness or contrast), API access (e.g., retrieving stock information via code as shown in ***Figure 1***), and edge detection (e.g., using edge detection operators from OpenCV).
>
> These emergent behaviors highlight the ***strong generalization*** of our tool-execution mechanism and demonstrate that DeepEyesV2 can ***independently learn to use new tools through RL*** without relying on tool-specific SFT data.
>
> 2. **Why RL can Achieve Adaptive Reasoning?**
>
> We observe that after RL training, DeepEyesV2 has developed the advanced reasoning capability of adaptive thinking. DeepEyesV2 no longer invokes tools for all questions, but instead makes adaptive choices based on the specific problem. We aim to analyze and explain why the use of only the simplest accuracy-based reward can enable DeepEyesV2 to acquire this capability.
>
> - **Essence of RL.** Many works (such as [1, 2, 3]) interpret the core of RL as ***improving sampling efficiency***. During RL, multiple trajectories are generated for a single problem, and training seeks to increase the likelihood of specific trajectories. In our experiments, for a given problem, DeepEyesV2 produces diverse trajectories: some do not invoke tools or use them sparingly, while others rely on frequent tool invocations.
>
> - **Simple Question.** For simple questions, correct answers can be obtained without invoking tools or with only minimal tool usage. In contrast, frequent or unnecessary tool invocations may introduce additional errors, ledding to low reward. Thus, for simple questions, RL increases the probability of trajectories that involve no or minimal tool usage, which is driven by the accuracy-based reward mechanism.
>
> - **Complex Question.** In contrast, for complex questions, minimal tool usage is insufficient to generate correct answers, and only complex tool invocation trajectories can potentially lead to rewards. Therefore, model still tends to adopt multi-turn tool usage for such scenarios.
>
> Thus, RL adjusts model’s preference for tool invocation across different scenarios, enabling DeepEyesV2 to acquire the capability of adaptive thinking.

---

> > ### Author Response · Authors · 2025-11-20
> > **General Response - 3 / 3**
> >
> > 3. **Accuracy of Tool Process during RL.**
> >
> > While using only an accuracy-based reward does not guarantee the correctness of tool usage during RL process, the vast majority of tool invocations by DeepEyesV2 are meaningful.
> >
> >
> > - **Unnecessary Tool Invocations.** We acknowledge that using only accuracy rewards may indeed lead to trajectories with unnecessary tool invocations yet correct results. This phenomenon is also prevalent in other works (such as DeepSeek R1), despite their strong reasoning capabilities, often generate trajectories with flawed reasoning processes but correct outcomes.
> >
> > - **Cold-start Ensures Correct Tool Use.** All tool invocations in our cold start data are meaningful and correct. Through SFT, model itself is less likely to generate meaningless tool calls.
> >
> >
> > - **RL Implicitly Mitigates Unnecessary Tool Calls.** As stated in ***our response to Q2***, excessive tool invocations can inherently introduce errors that prevent the model from obtaining rewards, prompting it to minimize tool usage as much as possible. Fewer tool invocations, in turn, reduce the likelihood of tool-related errors. Thus, our reward mechanism implicitly mitigates unnecessary tool calls.
> >
> >
> > - **Reward Hacking for Process Reward.** While process-based rewards on the model’s reasoning could, in principle, improve the correctness and logical consistency of tool usage, they introduce significant practical challenges and risks of reward hacking. A rule-based evaluator would require an unwieldy set of hand-crafted rules, making development and maintenance impractical. Using an LLM to score reasoning would demand additional preference data to train a reliable judge model and still leaves room for reward hacking. Additionally, similar conclusions are presented in DeepSeek-R1: it is challenging to evaluate and supervise the accuracy of the reasoning process. Given these trade-offs, we rely solely on an accuracy-based reward.
> >
> >
> >
> > In summary, while using only accuracy as the reward for RL may potentially lead to unnecessary tool invocations, our cold start data ensures the model avoids meaningless tool calls to a certain extent, and our RL framework can implicitly suppress such invocations.
> >
> >
> >
> >
> >
> > [1] Does Reinforcement Learning Really Incentivize Reasoning Capacity in LLMs Beyond the Base Model? NeurIPS 2025
> > [2] The Illusion of Thinking: Understanding the Strengths and Limitations of Reasoning Models via the Lens of Problem Complexity
> > [3] SFT Memorizes, RL Generalizes: A Comparative Study of Foundation Model Post-training

---

> > > ### Author Response · Authors · 2025-12-01
> > > **Further Analysis on Generalization of Tool Usage 1 / 2**
> > >
> > > We would like to further demonstrate the strong generalization capability of DeepEyesV2 in tool invocation. We conduct zero-shot testing of DeepEyes on TIRBench[1]. The tasks and required tool categories in TIRBench are not included in our cold-start and RL training data, therefore we think TIRBench can serve as an excellent platform for zero-shot testing of DeepEyesV2 to illustrate its generalization capability in tool invocation.
> > >
> > >
> > > 1. **Performance Comparison.**
> > >
> > > The performance of DeepEyesV2 on TIRBench is summarized in the following table. Specifically, DeepEyesV2 SFT represents the model after the cold-start stage, and DeepEyesV2 RL represents the model after RL training. It is evident that DeepEyesV2 substantially outperforms the baseline (Qwen2.5VL 7B) thanks to tool integration. This provides compelling evidence of DeepEyesV2's strong generalization ability, confirming that great performance improvements persist even when applied to **unseen tools and tasks**.
> > >
> > > | Model | Acc |
> > > | --- | --- |
> > > | Qwen2.5VL 7B | 16.0 |
> > > | DeepEyes | 17.3 |
> > > | DeepEyesV2 SFT | 18.7 |
> > > | DeepEyesV2 RL | 20.8 |
> > >
> > >
> > >
> > > 2. **New Tool Visualization.**
> > >
> > > We present examples of zero-shot tool utilization from TIRBench in Figure [1](https://imgur.com/ZNWjzmh
> > > ) and [2](https://imgur.com/CPsDZXW), comparing model performance before and after RL across tasks such as OCR, image rotation, and maze solving. We omit the specific options for each question and only display the core code segments. Prior to RL, lacking exposure to relevant tasks, model merely represent the input image. Conversely, the post-RL model demonstrates robust generalization capabilities. Specifically, for OCR task, DeepEyesV2 employs grayscaling and dilation to enhance character clarity, followed by cropping individual digits for recognition. For the rotation task, DeepEyesV2 rotates the image to determine its original orientation angle. Notably, these tasks and tools are absent from both our cold-start and RL datasets; yet, without additional training, DeepEyesV2 successfully comprehends and utilizes these novel tools. Furthermore, we highlight an interesting maze scenario where DeepEyesV2 generates code to simulate pathfinding, thereby verifying each option. Although the code contains minor imperfections, we think this strongly evidences DeepEyesV2's generalization potential with new tools. This comparison of tool usage pre- and post-RL effectively highlights DeepEyesV2's **exceptional adaptability to unseen tasks and tools**.
> > >
> > > 3. **Function Call Generalization.**
> > >
> > > Furthermore, beyond the code generalization capabilities, DeepEyesV2 demonstrates **strong generalization in function calling**. To evaluate this, we equip DeepEyesV2 with a rotation tool on a subset of the Rotated OCR task from TIRBench, requiring model to utilize the tool via function calls rather than by writing code. Since images in this task are rotated, model must perform rotation prior to OCR recognition. It is worth noting that in addition to the rotation function call, DeepEyesV2 retains the ability to invoke other tools via code generation. As shown in the table below, providing the rotation function call leads to further performance improvements, with DeepEyesV2 invoking the rotation tool in **15%** of instances. This demonstrates DeepEyesV2's strong generalization capability regarding newly added tools.
> > >
> > >
> > >
> > > | Model | Acc |
> > > | --- | --- |
> > > | DeepEyesV2 | 33.5 |
> > > | DeepEyesV2 w/ Rotate | 38.3 |
> > >
> > >
> > > We conducted **zero-shot testing** on a completely unseen benchmark to compare tool invocation patterns before and after RL. Complemented by additional function call experiments, these results fully demonstrate DeepEyesV2's **powerful tool generalization capabilities, encompassing not only the understanding of unseen tools but also proficiency in function calling.**
> > >
> > >
> > >
> > > [1] TIR-Bench: A Comprehensive Benchmark for Agentic Thinking-with-Images Reasoning

---

> > > > ### Author Response · Authors · 2025-12-01
> > > > **Further Analysis on Generalization of Tool Usage 2 / 2**
> > > >
> > > > We provide further elaboration on DeepEyesV2's tool generalization capabilities. Specifically, we categorize this generalization into three types:
> > > >
> > > >
> > > > 1. **Fixed System Prompt and Novel Tool.**
> > > >
> > > > First, using a fixed system prompt, DeepEyesV2 can **directly and correctly invoke entirely new tools** to solve unseen tasks. This capability is fully demonstrated by the performance comparison table and the figure illustrating tool invocation examples presented above.
> > > >
> > > >
> > > >
> > > > 2. **Generalization to New Function Call.**
> > > >
> > > > Second, regarding additional new function-based tools, we modify the system prompt to introduce these tools. DeepEyesV2 is able to correctly invoke them, resulting in performance improvements. This demonstrates that DeepEyesV2 also possesses **strong generalization capabilities for tools presented in a function format**.
> > > >
> > > >
> > > > 3. **Data Efficiency.**
> > > >
> > > > Finally, we address generalization regarding training data. Unlike the original DeepEyes, which was restricted to invoking a single tool, DeepEyesV2 employs code to call a diverse range of tools. Yet, DeepEyesV2 is trained on a dataset significantly smaller than that of DeepEyes. We think this data efficiency further underscores DeepEyesV2's robust generalization capabilities.
> > > >
> > > >
> > > > In summary, DeepEyesV2 demonstrates powerful tool generalization capabilities. (1) First, we execute tools through ***code*** rather than function calls, which endows DeepEyesV2 with strong diversification and extensibility. (2) Second, we conduct statistical analysis on the ***distribution*** of tool usage, showcasing the diversity of DeepEyesV2's tool invocations. (3) Third, through ***case studies***, we provide examples of DeepEyesV2's tool invocation generalization. DeepEyesV2 can learn new tools through RL without any cold-start data, such as calling stock APIs, rotation operations, maze navigation, and more. (4) Finally, we conduct further ***zero-shot testing*** on unseen tasks and tool categories, where DeepEyes demonstrated remarkable generalization capabilities. All of these fully demonstrate DeepEyesV2's powerful tool generalization abilities.

---

### Meta-Review · Area_Chair_yKPK · 2026-01-04

**Summary:**

The overall ratings gave by the reviewers are mixed, with one accept (8), two marginally below (4), and one marginally above (6).

Reviewers generally acknowledged the strong performance of DeepEyes V2, as well as the thoroughness of the analysis (e.g., the behavioral study of tool distribution and adaptive invocation).

Reviewers UjKn and zneT have concerns about the limited novelty of the two-stage training paradigm. Although the authors clarified the core novelty is “being the first to develop a multimodal agent model that organically integrates code execution and search”, the reviewers’ concern remain valid.

**Reviewer Concerns:**

Reviewers UjKn and zneT pointed out the insufficient analysis of what RL learns and how RL fosters “adaptive efficiency”.  The authors referred to representative cases in the manuscript and provided additional explanations. They promised revisions in the final manuscript, but it is unclear whether the reviewers concerns are fully addressed.

Other questions from the reviewers have been addressed by the authors.

**Reviewer Scores:**

Overall, the reviewers scores would likely remain the same if a full discussion period has occurred.

There is a slight chance that Reviewer UjKn might increase Rating from 4 to 5 given the extra experiment results on TIRBench and examples demonstrating the adaptive behavior and generalization of the model provided by the authors.

---

### Decision · Program_Chairs · 2026-01-26

Accept (Poster)